# Characterization of the planar differential mobility analyzer (DMA P5) : resolving power, transmission efficiency and its application to atmospheric relevant cluster measurements

Zhengning Xu[1], Jian Gao[1], Zhuanghao Xu[1], Michel Attoui[3], Xiangyu Pei[1], Mario Amo-González[4], Kewei Zhang[1], Zhibin Wang[1,2*]

[1]Zhejiang Provincial Key Laboratory of Organic Pollution Process and Control, MOE Key Laboratory of Environment Remediation and Ecological Health, College of Environmental and Resource Sciences, Zhejiang University, Hangzhou 310058, China

[2]ZJU-Hangzhou Global Scientific and Technological Innovation Center, Zhejiang University, Hangzhou 311215, China

[3]University Paris-Est Creteil, University Paris-Diderot, LISA, UMR CNRS 7583, France

[4]MION S.L., Avda. Francisco Valles 8, Boecillo, Valladolid, 47151, Spain

*Correspondence to*: Zhibin Wang (wangzhibin@zju.edu.cn)

**Abstract.** The planar differential mobility analyzer (DMA), functioning as a particle sizer, exhibits superior transmission and selection accuracy at ambient pressure relative to its cylindrical counterparts. It also presents integration potential with atmospheric pressure interface mass spectrometry (API-MS) for enhanced cluster detection with an additional ion mobility dimension. In this study, the performance of a commercially available planar DMA (DMA P5) was evaluated. The device is capable of sizing particles below 3.9 nm, with larger sizes measurable through a sheath gas flow restrictor. The resolving power was appraised under various recirculation arrangements, including suction and counter-flow modes, along with different sheath flow rates, using electrosprayed tetra-alkyl ammonium salts. The peak resolving powers for tetrahexylammonium (THA[+]) achieved in suction and counter-flow modes were 61.6 and 84.6, respectively. The DMA P5 offers a sizing resolution that is 5 to 16 times greater than that of cylindrical DMAs. Resolving power displayed a near-linear relationship with the square root of the applied voltage ($\sqrt{V_{DMA}}$) in counter-flow mode. Conversely, the resolving power for THA[+] ceased its linear enhancement with $\sqrt{V_{DMA}}$ beyond $V_{DMA}$ of 3554.3 V, entering a plateau, which is ascribed to the perturbations in sample flow impacting the laminar nature of sheath flow. The DMA P5 transmission efficiency reaches 54.3%, markedly surpassing that of conventional DMAs by nearly an order of magnitude. Moreover, the mobility spectrum of various electrosprayed tetra-alkyl ammonium salts, along with the mass-to-charge versus mobility two-dimensional spectrum of sulfuric acid clusters, were characterized using the DMA P5-MS system.

## 1 Introduction

Measurement of the physical and chemical properties of particles under 3 nm is crucial for elucidating the mechanisms behind atmospheric aerosol nucleation (Kerminen et al., 2018). The Differential Mobility Analyzers (DMA) has been a staple for

sizing and classifying aerosols (Knutson & Whitby, 1975). Specifically, the TSI Nano-DMA (Model 3085, Chen et al., 1998) and the TSI 1 nm-DMA (Model 3086, Stolzenburg et al., 2018) are engineered to size particles down to 3 nm and 1 nm, respectively. When integrated with the DEG-based scanning mobility particle sizer (SMPS) system (Jiang et al., 2011), these DMAs facilitate the measurement of size distributions for sub-3 nm particles (Cai et al., 2017).

A variety of mass spectrometry-based methodologies have been employed to ascertain the chemical composition of clusters and their precursors (Chen et al., 2020; Peng et al., 2022). Nonetheless, these methods do not provide direct measurements of cluster size and structure. In the Thermal Desorption Chemical Ionization Mass Spectrometer (TDCIMS) system (Smith et al., 2004), size-resolved particles are segregated using DMAs and subsequently analyzed by mass spectrometry after sufficient accumulation via electrostatic deposition. Presently, the detection limit of TDCIMS is established at 5 nm (Perraud et al.,

2020). Studies on clusters necessitate a substantially extended accumulation period, owing to the precipitous decline in charging efficiency as particle diameter decreases, coupled with the low ion transmission efficiency of the deployed DMAs. Integration of DMAs with mass spectrometers (DMA-MS) allows for the immediate detection of ion mobility and chemical composition, showing considerable promise as a tool for investigating the physicochemical characteristics of atmospheric clusters, as well as the mechanisms underlying particle nucleation and initial growth (Zhang et al., 2022). The sole constraint

of this technique is the performance of the DMA in sizing particles below 3 nm.

Planar DMAs have been effectively integrated with a range of commercial Atmospheric Pressure Interface Mass Spectrometers (API-MS), achieving high resolution and high transmission efficiencies (Hogan and Fernandez de la Mora, 2009; 2010; Hogan et al., 2011; Criado-Hidalgo et al., 2013). Using DMA-MS, the physicochemical properties of various atmospheric clusters have been elucidated, including metal iodide clusters (Oberreit et al., 2014; 2015), complexes of DMA with sulfuric acid

(Ouyang et al., 2015; Thomas et al., 2016), sodium chloride clusters (Li and Hogan, 2017), and hybrid iodine pentoxide–iodic acid clusters (Ahonen et al., 2019). Rus et al. (2010) characterized several prototypes of planar DMAs, revealing that the transmission efficiency of the DMA P4 model was approximately 50%. Improving upon the DMA P4, the DMA P5 iteration refined the design of the outlet to ensure that coupling and decoupling the DMA with/from an MS would not compromise the vacuum integrity (Amo-González and Pérez, 2018).

The characterization of the commercial DMA P5 remains unreported. This study characterizes the performance of DMA P5 under a variety of operational conditions. Resolving power and transmission efficiency were assessed using standard ion clusters. Additionally, DMA P5 was utilized to determine the ion mobility spectrum of electrosprayed clusters from different tetra-alkyl ammonium salts and to generate the mass-to-charge ratio versus ion mobility two-dimensional spectrum for sulfuric acid.

**2 Experimental setup and methodology**

In this study, a DMA P5 (SEADM, Valladolid, Spain) was employed, comprising a pair of parallel plates, separated by insulating holders. This configuration facilitates a spatial mobility filter through the interplay of horizontal laminar sheath flow

and a vertical electric field in the electrode separation region (Purves et al., 1998). Polydispersed aerosols entering via an inlet slit are directed towards the outlet electrode by the electric field. For specified sheath flow velocities and electric field intensities, monodispersed aerosols within an exceedingly narrow ion mobility range exit through the outlet orifice and are conveyed to detectors. The intricate design and precise dimensions of the DMA P5 are delineated by Amo-González and Pérez (2018). The theoretical underpinnings of particle sizing using planar DMA are elaborated in the Supplementary Information (Section 1). The selected ion mobility can be calculated using the following equation:

$$Z = \frac{U \cdot h^2}{L \cdot V_{DMA}} \qquad (1)$$

where, $Z$ denotes the ion mobility under selection, $U$ signifies the velocity of sheath flow along the symmetry plane bisecting the inlet slit, $h$ is the inter-electrode distance, $L$ indicates the horizontal displacement between the inlet slit and the outlet orifice, and $V_{DMA}$ represents the potential difference across the electrodes.

The recirculation system designed for the DMA P5 apparatus is required to deliver a particle-free sheath flow with consistent velocity and temperature. In the current configuration, the system incorporates a high-capacity air blower (Ref 497.3.265-361, Domel), a water cooler integrated with a constant temperature water bath (DCW-2008, SCIENTZ), a high-velocity-compatible particle filter, and NW40 and NW50 corrugated stainless-steel tubing with corresponding connectors. The filtration unit includes a flat-panel commercial HEPA filter (Ref 34230010, Megalem MD143P3, Camfil Farr) and a two-part stainless-steel housing. The upper portion of the housing consists of an NW40 connector, and the lower part is designed to match the planar HEPA filter geometry. An O-ring and screws secure the HEPA filter, clamped between the two lower faces of the housing. All components were sourced from local suppliers. Substitute components with equivalent specifications should not compromise system functionality. The blower rotational speed is modulated by a 0-10 Vdc analogue signal ($V_{blower}$). The flow rate for different control voltages was determined using Equation (1), based on the ion mobility and $V_{DMA}$ of a standard ion (THA$^+$) (Fig. S2). The system maintains a controlled temperature of 24 degrees Celsius.

The nano-ESI ionization source, procured from SEADM, was adapted for this application. The core of the nano-ESI ionization source is composed of a PEEK cubic electrospray chamber, which at its base houses the inlet electrode of the DMA P5. It is hermetically sealed at both the front and back by dual glass windows and features a centrally positioned 1/8" capillary guide at the top. Additionally, the chamber is equipped with two 1/4" ports on the upper left and right sides for the purpose of flow injection and exhaust, respectively.

In the current study, two recirculation circuit configurations—suction mode and counter-flow mode—were examined in relation to the design of the ionization source. In suction mode, aerosols are directed towards the inlet slit by a synergistic effect of the electric field established between the electrospray and the DMA inlet electrode, coupled with the polydisperse aerosol flow ($Q_{in}$). These aerosols are then drawn into the separation region by the monodisperse aerosol flow ($Q_{out}$), while the surplus flow ($Q_{excess}$) is expelled through an alternative port. In contrast, the counter-flow mode employs a single port to discharge the counter-flow ($Q_{count}$) away from the inlet slit. A compensating flow ($Q_c$), with a flow rate equivalent to the sum

of $Q_{count}$ and $Q_{out}$, is introduced into the system via a T-connector. The specifics of these recirculation circuit arrangements are depicted in Fig. 1.

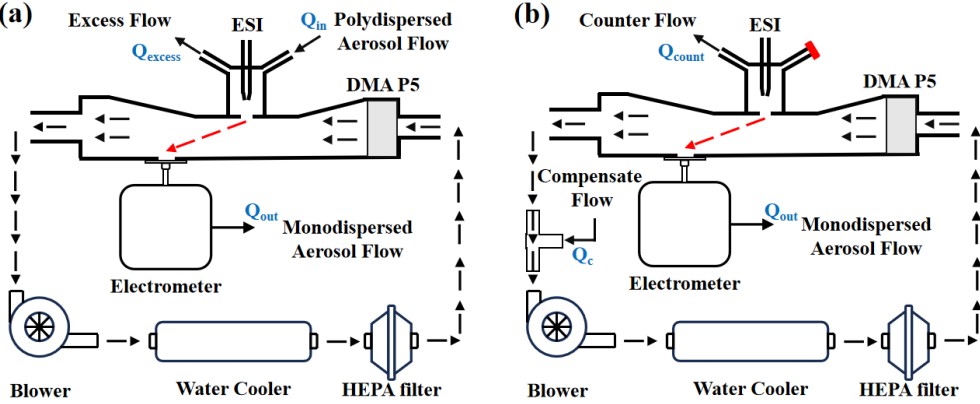

**Fig.1 Schematic diagram of the recirculation circuit setups (a) suction mode; (b) counter-flow mode of DMA P5 system.**

The operational parameters of the DMA P5 were assessed through the classification of aerosols produced by electrospraying

salt solutions. A silica capillary (FS360-50-N-5-C50, New Objective) was installed within the nano-ESI ionization source capillary guide to facilitate the connection between the ionization chamber and the solution vial. The capillary was secured at both ends using standard liquid chromatography fittings. A T-connector linked the vial to a pressure sensor and a valve-equipped syringe. The propulsion of the solution through the capillary was achieved by pressurizing the gas within the vial to reach the target pressure over the solution. Following pressurization, the valve preceding the syringe was closed to maintain

consistent pressure. An inert metal wire, submerged in the solution, was utilized to introduce a high voltage to the system. This approach aligns with protocols outlined by Jiang et al. (2011) and Cai et al. (2018). By applying a positive voltage and omitting subsequent neutralization steps, it is possible to produce singly-charged aerosols with a positive polarity (Ude and Fernández de la Mora 2005). For this experiment, the electrospray source pressure ($P_{ES}$) and voltage ($V_{ES}$) were set to approximately 20 kPa and 2000 V, respectively. Particle counts were measured using Faraday cage electrometers (Lynx E11&E12, SEADM,

Valladolid, Spain, Fernandez de la Mora et al., 2017), with an output signal range of 0-2V and amplification factors of $10^{11}$ V/A and $10^{12}$ V/A, respectively. Ion mobility for any aerosol with a known DMA voltage was calibrated against standard ions under a constant sheath flow velocity, according to the formula in Eq. (2).

$$Z = \frac{V_{standard}*Z_{standard}}{V_{DMA}} \qquad (2)$$

where, $Z_{standard}$ denotes the electrical mobility of a benchmark ion, while $V_{DMA}$ and $V_{standard}$ refer to the voltages across the DMA

for the targeted aerosol and the benchmark ion, respectively, under equivalent sheath flow conditions. THA$^+$, a standard ion with a well-characterized mobility of 0.97 cm²V⁻¹s⁻¹, as reported by Ude and Fernández de la Mora (2005), was produced through positive electrospraying of a 0.5 mM THABr solution in a 9:1 methanol/water mixture. This study calculates the mobility diameter employing the Stokes-Cunningham equation, as delineated by Tammet (1995) and Wiedensohler et al.

(2012). Comprehensive details regarding the computation of mobility diameter are presented in the Supplementary Information (Section 2). The DMA P5 particle sizing capability, within the current setup and subject to a maximum $V_{DMA}$ of 10 kV, spans from 1.9 to 3.9 nm. Nevertheless, incorporation of a flow restrictor within the sheath gas circuit permits the extension of the upper sizing limit. It was observed that an increased sheath flow rate inversely affects the particle sizing range, as depicted in Figure S2.

The sizing resolution ($R$) of a particular aerosol is quantitatively determined by the ratio of its central electrical mobility (Z) to the full width at half maximum (FWHM) of its peak ($\Delta Z_{FWHM}$) , according to Flagan (1999):

$$R = \frac{Z}{\Delta Z_{FWHM}} \quad (3)$$

The theoretical calculation of planar DMA sizing resolution is given by the following equation (Supplementary Information Section 3 for detailed derivation process):

$$R^{-1} = \frac{\Delta Z_{FWHM}}{Z} = \sqrt{\left[\left(\frac{Q_{in}+Q_{out}}{L_{slit} 2\,Re\,v}\right)^2 + \frac{16ln2kT}{V_{DMA}N_e}\left(1 + \left[\frac{h}{L}\right]^2\right)\right]} \quad (4)$$

where, $Q_{in}$ represent the polydispersed aerosol flow rate and $Q_{out}$ is the monodispersed aerosol flow rate, respectively, $L_{slit}$ is the length of inlet slit, $Re$ is the Reynolds number, $v$ is the viscosity of the sheath gas, $k$ is the Boltzmann's constant, $T$ is the absolute temperature of the sheath gas, $Ne$ is the net charge on the aerosol.

## 3 Results and Discussion

### 3.1 Resolving Power under different recirculation modes

The application of an ESI source in conjunction with a DMA P5 necessitates consideration that the ESI voltage is variable in scan mode. The nano-ESI source voltage is maintained at a differential above the DMA P5 inlet electrode voltage, the differential voltage corresponds to the actual ESI voltage. This configuration mitigates the effects of voltage fluctuations during scan mode on electrospray formation and stability. Additionally, the electric field established between the capillary tip and the inlet electrode channels aerosols into the DMA separation chamber effectively. As previously detailed, DMA P5 can operate in two recirculation modes to evaluate its performance with aerosols generated from the nano-ESI source. In suction mode, polydisperse aerosols are drawn into the separation zone of the DMA. In contrast, the counter flow mode applies an electrical insertion of aerosols into the DMA. The inlet slit counter flow enhances droplet evaporation and prevents neutral droplets from accessing the separation zone.

Based on Eq. (4), large $Re$ is expected in order to achieve high sizing resolution, under which condition the sheath flow velocity needs to be very large based on the following equation:

$$Re = \frac{Uh}{v} \quad (5)$$

The aspiration of an aerosol sample can perturb the sheath flow, thereby diminishing the sizing resolution. Incomplete desolvation of the solvent and the intrusion of neutral droplets into the DMA separation region during aspiration may alter the

peak shape, thus complicating the identification of targeted analytes (Amo-González and Fernández de la Mora, 2017).

Conversely, while the suction mode may yield a higher aerosol number concentration, this increase comes at the cost of reduced resolving power and degraded peak morphology.

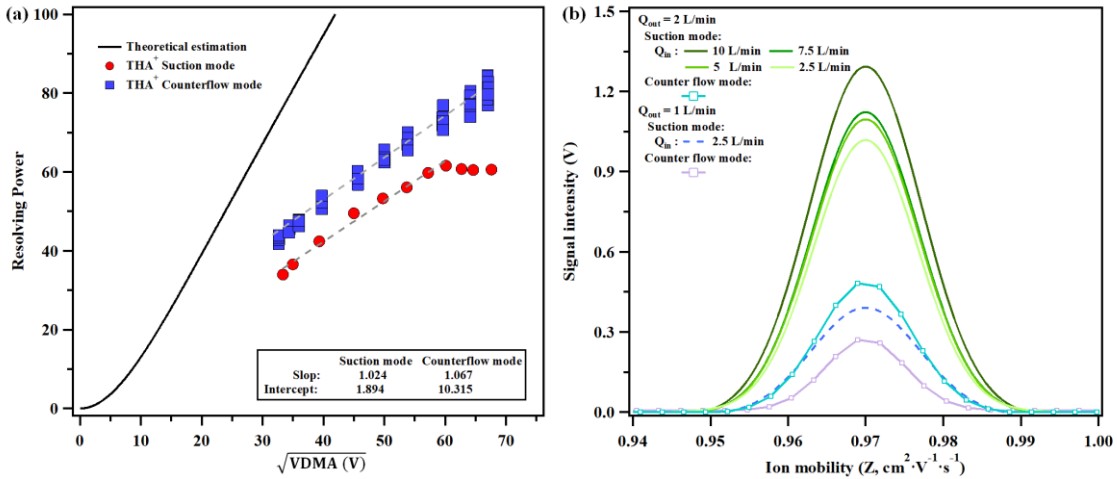

**Fig 2. (a) The dependency of the resolving power of THA$^+$ on DMA voltage ($V_{DMA}$) under suction mode and counter flow mode; (b) dependency of the resolving power and signal intensity on the $Q_{out}$ under suction mode and the comparison with counterflow mode.**


Equation (1) suggests that $V_{DMA}$ exhibits an approximately linear relationship with sheath gas velocity when ion mobility is held constant. As per Equation (4), the resolution ($R$) escalates as $\sqrt{V_{DMA}}$ increases, given a defined DMA structure (height ($h$) and length ($L$)), sheath flow characteristics (kinematic viscosity ($v$), temperature ($T$), and flow rate ($U$)), and the aerosol charge state ($N_e$). The efficacy of DMA P5 in capturing the THA$^+$ mobility spectrum via suction mode is depicted in Figure S3. For

solution concentrations exceeding 0.5 mM, distinct THA$^+$ monomer, dimer, and trimer peaks are discernible. Elevated solution concentrations enhance the signal-to-noise ratio but inversely affect the dimer-to-monomer ratio. Equations (1) and (4) infer that both $V_{DMA}$ and the associated sizing resolution for THA$^+$ are augmented with increasing sheath flow rate. Figure 2a illustrates the resolving power dependence on $\sqrt{V_{DMA}}$ for sizing 1.47 nm ions (THA$^+$) under both recirculation modes, modulated by varying the sheath flow velocity. The inlet flow rate ($Q_{in}$) is set at 5 L/min in suction mode, while the counterflow

rate ($Q_{count}$) is maintained at 0.5 L/min in counterflow mode.

In suction mode, a positive linear correlation between $R$ and the $\sqrt{V_{DMA}}$ was observed, characterized by a slope of 1.024 and an intercept of 1.894 when $V_{DMA}$ was below 3554.3 V. Beyond this voltage threshold, $R$ remained essentially constant despite increases in $\sqrt{V_{DMA}}$. Specifically, in the high $V_{DMA}$ range, which corresponds to higher sheath flow velocities, $R$ was confined within 60 ± 3. Conversely, under counter-flow mode, a robust linear relationship ($R^2 = 0.95$) between $R$ and $\sqrt{V_{DMA}}$ was

demonstrated across the operational sheath flow velocity range of the system, with a slope and intercept of 1.067 and 10.315, respectively.

The resolving power of THA$^+$ typically exhibited higher values in counter-flow mode, closely aligning with theoretical predictions. As sheath flow velocity increased, the disparity between measured and theoretical resolving power progressively widened. In suction mode, the presence of a polydispersed sample flow resulted in resolving power being reduced by 14.9% - 21.7% compared to counter-flow mode for $\sqrt{V_{DMA}}$ values less than 59.6. This discrepancy escalated to 24.9% as $\sqrt{V_{DMA}}$ reached 66.0. The divergences in resolving power between the two recirculation modes, and the deviations from theoretical calculations, can be primarily ascribed to turbulence effects. At higher sheath flow velocities, the flow is more prone to turbulence, which is corroborated by the observation that resolving power ceased to increase at $V_{DMA}$ values exceeding 3554.3V. The growing gap between measured and theoretical resolving power can be attributed to the sheath flow insufficient stabilization under laminar conditions. The introduction of an additional set of pre-laminarizer screens in the sheath flow recirculation circuit under counter-flow mode has been shown to enhance resolving power, bringing it closer to the theoretical maximum (Amo-González and Pérez, 2018).

Figure 2b illustrates the relationship between the peak signal intensity of THA$^+$ and its resolving power as a function of various inlet flow rates ($Q_{in}$) under suction mode. The results indicate that variations in $Q_{in}$ exert a minimal impact on the resolving power, which ranged between 58.06 and 61.58, corresponding to a relative deviation from -3.5% to 2.3%. Conversely, the signal intensity, which reflects the number concentration of the sizing aerosol, is significantly affected by changes in $Q_{in}$. With a constant outlet flow rate ($Q_{out}$) of 2 L/min, the peak signal intensities for THA$^+$ were measured at 1.31 V, 1.13 V, 1.1 V, and 1.02 V for $Q_{in}$ values of 10 L/min, 7.5 L/min, 5 L/min, and 2.5 L/min, respectively. A notable reduction in signal strength was observed when $Q_{out}$ was lowered to 1.0 L/min. It is also worth mentioning that an increase in signal intensity from the electrometer was achieved in suction mode at the expense of decreased resolving power. In comparison to counter-flow mode, the peak signal intensity of THA$^+$ in suction mode was significantly higher, ranging from 108.2% to 167.3% for the same $Q_{out}$. In the counter-flow mode, two parameters can). Figures 3a and b exhibit the variations in the central voltage and signal intensity of THA$^+$ under different $Q_{out}$ values at $Q_{count}$ settings of 0.5 L/min and 1.0 L/min, respectively. The signal strength generally exhibits a monotonic increase with $Q_{out}$. In accordance with Knutson & Whitby (1975), the central voltage of THA$^+$ inversely correlates with $Q_{out}$. $Q_{out}$ exerts a more pronounced effect on measurement outcomes compared to $Q_{count}$. Figure 3c depicts the correlation between the integrated peak area of THA$^+$, resolving power, and central voltage as a function of $Q_{out}$. As $Q_{out}$ rises from 0.5 L/min to 3.0 L/min, there is a decrement in the central voltage of THA$^+$ by approximately 20 V (under 1%), whereas the integrated peak area surges by factors of 7.6 and 6.8 at $Q_{count}$ of 1.0 L/min and 0.5 L/min, respectively, and the resolving power diminishes from 75 to 69. The counting efficiency of the electrometer is defined as the integrated peak area divided by $Q_{out}$. The trend depicted in Fig. S4 indicates that counting efficiency increases with $Q_{out}$ up to 2 L/min and declines thereafter. Therefore, setting a total $Q_{out}$ above 2.0 L/min is deemed inadvisable due to the resultant compromise in both counting efficiency and resolving power, particularly when interfaced with an API-MS.

It is pertinent to note that the recirculation setups described above are designed for the analysis of aerosols originating from an ESI source. For atmospheric cluster measurements, Secondary Electrospray Ionization (SESI) (Rioseras et al., 2017) is employed, wherein reagent ions are generated through the electrospraying of selectively chosen solutions. In the injection

mode, clusters enter the DMA via the polydisperse aerosol flow inlet. In the counter-flow mode, the obstructed port, marked in red in Figure 1b, serves as the entry point for atmospheric clusters. Here, the overall counter flow ($O_{count}$) equates to the aggregate of the counter-flow rate and the sample flow rate. Gao et al. (2023) have utilized the SESI-DMA-Time of Flight (TOF) MS setup to quantify the products resulting from α-pinene ozonolysis.

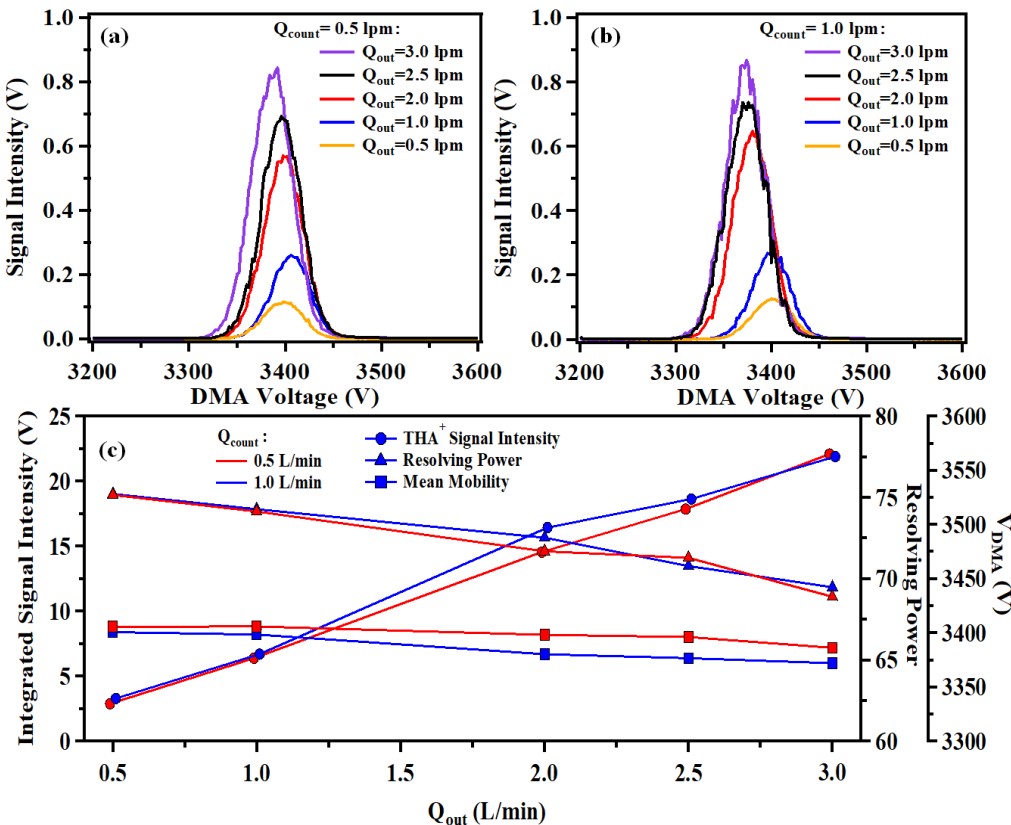

**Fig. 3** Mobility spectrum of THA$^+$ under different $Q_{out}$ with (a) $Q_{count}$ = 0.5 L/min; (b) $Q_{count}$ = 1.0 L/min; (c) Integrated signal intensity, resolving power and $V_{DMA}$ of THA$^+$ under different $Q_{out}$.

The size resolution of the THA$^+$ monomer, determined using the DMA P5 and the Half Mini DMA (Fernández de la Mora and Kozlowski, 2013), was assessed in our laboratory and compared with the outcomes reported for various commercial DMAs (Jiang et al., 2011; Stolzenburg et al., 2018). The DMA P5 was operated in counter-flow mode at a sheath flow rate of approximately 1500 L/min, which corresponds to a $V_{blower}$ of 8.5 V. The Half Mini DMA functioned at an aerosol-to-sheath flow ratio of 10/300 L/min. Resolutions previously reported were obtained at aerosol-to-sheath flow ratios of 0.6/6 L/min for the Caltech nanoRDMA, 6/61.4 L/min for the Vienna DMA, 2/21.9 L/min for the Grimm nanoDMA, 2.0/20 L/min for the TSI 3085, 2.5/25 L/min for the TSI 3086, and 1.5/15 L/min for the Caltech RDMA. For all referenced cylindrical DMAs (excluding

the HalfMini DMA), the typical aerosol-to-sheath flow ratio is approximately 10, which is the standard configuration for particle sizing in both laboratory and field studies. The comparative analysis reveals that the planar DMA exhibits superior sizing resolution, registering 5-16 times higher than that of cylindrical DMAs (Fig. 4). However, the high costs associated with maintaining the elevated sheath flow rates render the DMA P5 impractical for atmospheric particle number size distribution

measurements. Conversely, the high resolution and ion transmission efficiency are nearly synonymous with planar DMAs. This distinct advantage warrants further exploration, particularly for enhancing cluster detection in conjunction with a mass spectrometer, thereby adding an additional ion mobility dimension. Figure S5 presents the mobility spectrum of THA$^+$ acquired using the Half Mini DMA. The distinctly separated THA$^+$ ion beam thus obtained was subsequently employed to characterize the ion transmission efficiency of the DMA P5.

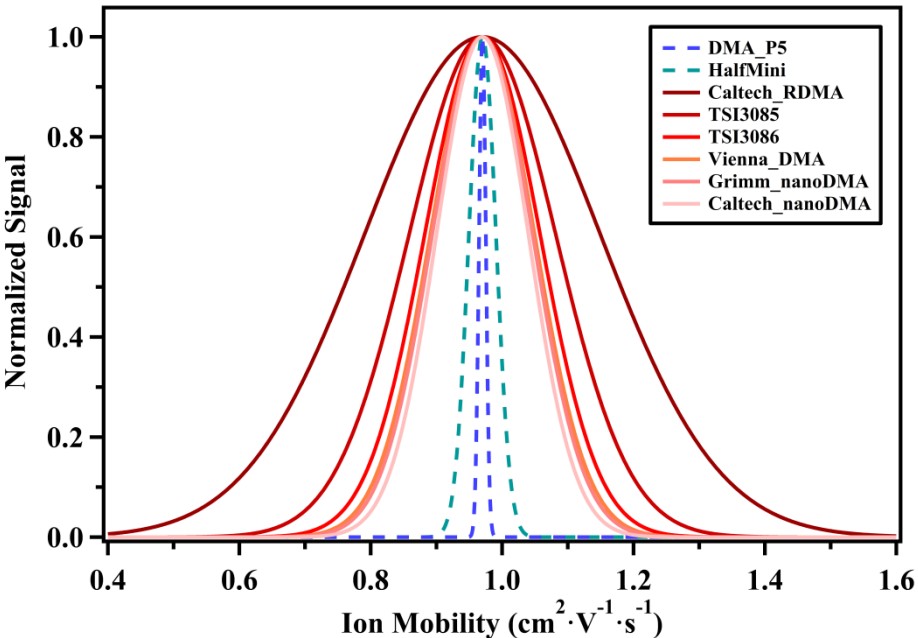


**Fig. 4 Comparison of the resolving power of DMA P5 and other commercial DMAs for detecting THA$^+$. DMA P5 was operated under the sheath flow rate of 1500 L/min, Half Mini was operated under the sheath flow rate of 300 L/min. All the signal intensiy is normalized with the peak height.**

In our investigation, the DMA P5's sizing capabilities were extended beyond THAB to include three additional tetraalkyl ammonium halides: tetra methyl ammonium iodide (TMAI), tetra butyl ammonium iodide (TBAI), and tetra decyl ammonium bromide (TDAB), each at a concentration of 0.5 mM/L. Ion mobility spectra for both positive and negative ions are illustrated in Figures 5 and 6. The spectra indicate that the solutions yield well-defined monomers and dimers with a single positive charge. Additionally, trimers were observed in solutions of TBAI and THAB. In the negative ion spectra, halogen elements

predominantly emerge as negative ions. The mobilities of aerosol ions were calculated using Equation (2), with the findings detailed in Tables S1 and S2. The calculated mobilities align with the values previously reported by Ude et al. (2005).

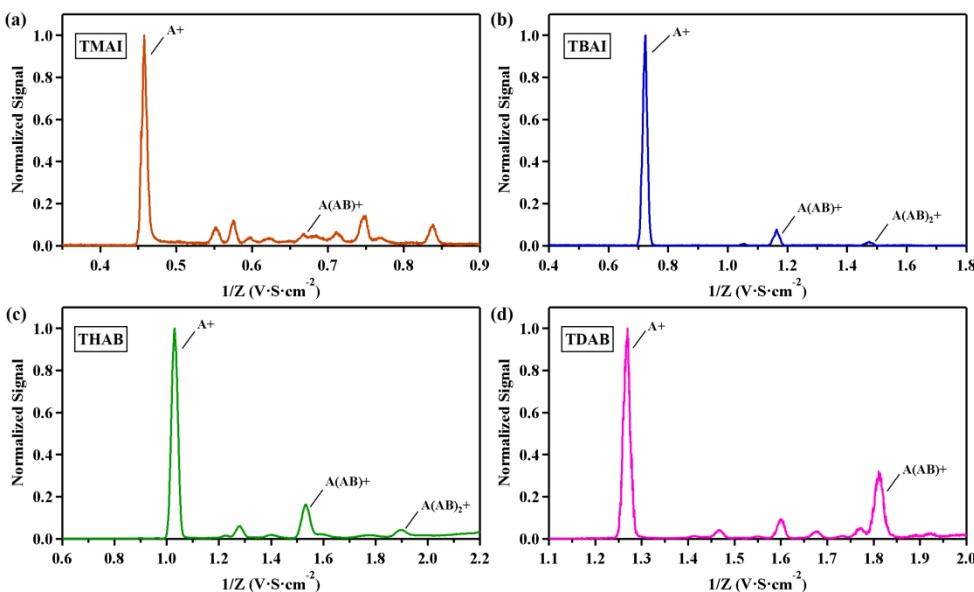

**Fig. 5 Positive mobility spectrum of electrosprayed tetra-alkyl ammonium ions, (a) TMAI, (b) TBAI, (c) THAB, (d) TDAB.**

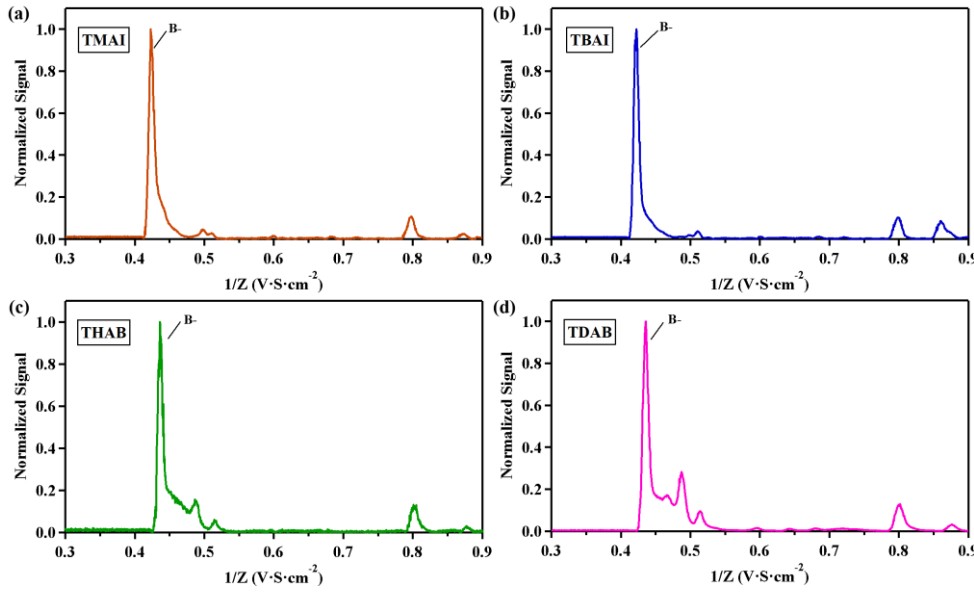

**Fig. 6 Negative mobility spectrum of electrosprayed tetra-alkyl ammonium ions, (a) TMAI, (b) TBAI, (c) THAB, (d) TDAB.**

As delineated by Equation (4), the resolving power of the planar DMA is a function of $V_{DMA}$ and $Ne$. Figure 7 elucidates the relationship between the resolving power of positively charged monomers, including THAB, TMAI, TBAI, and TDAB, and

$V_{DMA}$ across varying sheath flow velocities. The analysis revealed a strong linear correlation ($R^2 = 0.96$) between R and $\sqrt{V_{DMA}}$ for the various tetraalkyl ammonium cations, characterized by a slope of 0.992 and an intercept of 13.511. As depicted in Fig. 7, a majority of the data points are encapsulated within the 95% confidence interval. These findings provide a reference for predicting changes in the resolving power of DMA P5 operating in counter-flow mode at different sheath flow rates. Additionally, the resolution function may be utilized for multi-peak fitting, facilitating the identification of peaks within the ion mobility spectrum of multi-component complexes.

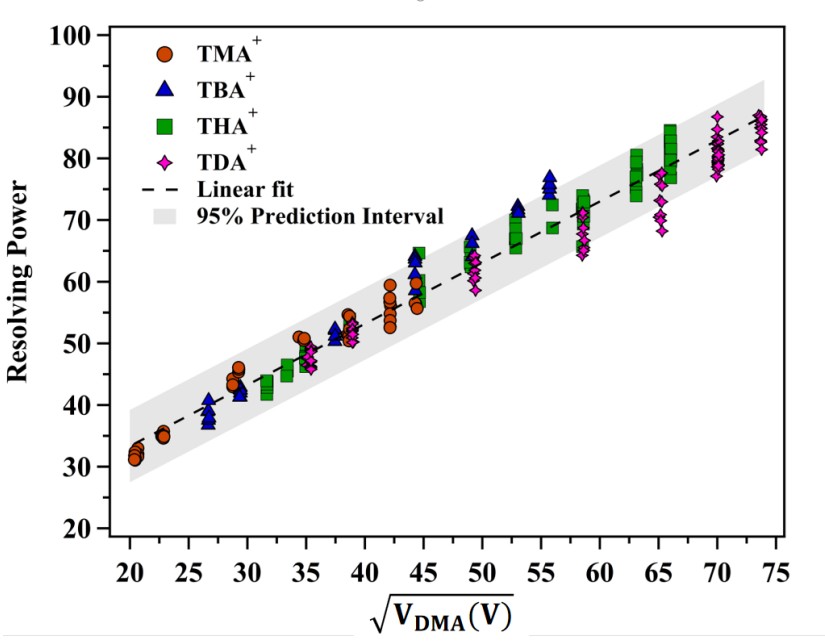

**Fig. 7 Relation of resolving power of different cluster ions with $V_{DMA}$**

## 3.2 Transmission efficiency

The pivotal consideration in employing the DMA P5 for atmospheric cluster studies is its ability to isolate ionized clusters in quantities sufficient for subsequent analysis within an acceptable timeframe. Thus, it is imperative to characterize the DMA P5's performance, particularly its transmission efficiency. Studies incorporating electrospray sources positioned directly opposite a planar DMA's inlet slit have documented significant increases in ion transmission compared to cylindrical DMAs. Javaheri et al. (2008) demonstrated the potential for near-total transmission of molecules from a dilute electrosprayed solution as ions to the atmospheric inlet of a mass spectrometer. Similarly, Fernandez de la Mora (2019) reported the transmission of over 1 nA of dominant electrosprayed ions to the outlet of a planar DMA. Such efficiency has been leveraged by Tauber et al. (2018) to introduce DMA-refined, highly concentrated atomic ions into a nucleation chamber for the investigation of ion-induced nucleation. This exceptional level of transmission is unattainable with cylindrical DMAs and underpins the prevalence

of planar DMAs in successful integrations with mass spectrometry (MS) systems, predominantly those utilizing electrospray sources. For instance, Fernandez de la Mora et al. (2005) noted that it took in excess of one hour to accrue a single mass spectrum containing relevant data on $(THABr)_n(THA^+)_z$ clusters using a high-resolution cylindrical DMA with an aerosol flow of 10 L/min at a constant voltage in conjunction with a Time of Flight (TOF) MS. However, there are exceptions, as Ude et al. (2004) demonstrated with quadrupole MS systems fixed on a single mass while a cylindrical DMA scanned across voltages. In addition, Steiner et al. (2014) successfully coupled a high-resolution cylindrical DMA with a TOF-MS to study small ions emanating from radioactive bipolar sources.

Electrosprays represent intense unipolar ion sources, injecting currents of several hundred nanoamperes (nA) into areas typically under 1 μm². In such scenarios, ion loss is predominantly due to beam broadening from space charge effects, a phenomenon generally not pertinent to most atmospheric measurements. The transmission efficiency of the DMA P5 was assessed using a tandem DMA configuration (refer to Figure S6). The majority of ions generated by electrospray are dissipated by space charge within the aerosol inlet tube and the annular space before the analyzing region's inlet slit of the first DMA. As a result, by the time the ions reach the subsequent DMA, the space charge effect is considerably reduced. Hence, the losses recorded are primarily attributed to diffusion, which aligns with atmospheric sampling. To isolate the $THA^+$ monomer, a Half Mini DMA operating at fixed sheath flow and voltage was utilized as the upstream DMA. This setup also serves to reduce the impact of multi-charged ions possessing higher molecular weights (Attoui et al., 2013).

Standard aerosol ions (specifically $THA^+$) were produced using the previously outlined method. Prior to each experiment, a comprehensive mobility spectrum was recorded in the voltage scanning mode to ensure that only $THA^+$ monomers were present, as confirmed by comparison with the published result (Liu et al., 2021). Two electrometers, Lynx E11 and E12, with amplification factors of $1\times10^{11}$ V/A and $1\times10^{12}$ V/A respectively, were employed to quantify the aerosol number concentration before and after the DMA P5. An ancillary experiment was conducted to ascertain the correction factor required, involving the connection of both electrometers to the Half Mini DMA via a flow splitter (TSI 3780) with uniform tube lengths. The Half Mini DMA functioned at a constant voltage to isolate $THA^+$ ions exclusively. Sample flow rates for the electrometers were determined using a bubble flowmeter, with signals from both electrometers recorded concurrently. The resultant correction factor, 10.34, aligns closely with the theoretical amplification ratio between the two electrometers. To compensate for ion losses to the tube walls, the lengths of tubing from the flow splitter (TSI 3780) to the electrometers, excluding the DMA P5, were identical.

Following the Half Mini DMA, the DMA P5 was situated downstream, interfaced between the flow splitter and the secondary electrometer. Connectivity was facilitated using a 3D-printed cubic chamber, designed with a cylindrical inner geometry featuring a diameter of 32 mm and a height of 6.5 mm. The inlet tube was inserted directly through a hole on the top of the chamber, aligning centrally within the cylindrical space. To ensure an airtight seal, the gap between the conductive silicone tube and the chamber hole was filled with silicone adhesive. The proximal end of the inlet tube was linked to the beam splitter, with its distal end terminating flush with the lower extremity of the chamber. Upon securing the cubic chamber to the inlet electrode of the DMA P5, a 5.2 mm gap was maintained between the distal end of the inlet tube and the DMA's inlet slit.

Throughout the experimental procedure, the voltage applied to the Half Mini DMA remained constant, whereas the voltage across the DMA P5 was varied progressively. The ion number concentration detected by both electrometers was calculated using the following equation:

$$N = V_{EM}/(amp \cdot e \cdot q) \qquad (6)$$

where $V_{EM}$ represent the signal of electrometer, $amp$ is the amplification coefficient of the electrometer, $q$ is the sample flow rate of electrometer and $e$ represents elementary charge. The recorded concentration was then corrected using the correction factor.

The DMA P5 was operated in suction mode, with the inlet electrode grounded and the outlet electrode connected to a negative voltage. A straight 10 cm length of dissipative plastic tubing (85A, FreelinWade, Oregon) provided an electrically stable connection between the outlet electrode and the grounded downstream electrometer, effectively mitigating electrostatic build-up on the tube interior surface (Attoui et al., 2016). The integrity of this setup was critical, ensuring that the dissipative tubing remained straight and in firm electrical contact with both the outlet electrode and electrometer. Owing to challenges in managing the requisite electric field, the counter-flow mode was not evaluated. The conventional configuration, which employs a positive voltage at the inlet electrode and grounding the outlet electrode, could not effectively propel ions against the counter flow, particularly with the sample tube in place. Experiments were conducted across varying sheath flow velocities, each corresponding to distinct control voltages as delineated in Figure S2.

The transmission illustrated is the maximum transmission efficiency refers to the peak value of the aerosol number ratio counted by the downstream to upstream electrometers ($N_{down}/N_{up}$). This metric is depicted as the height of the peak in the ratio distribution. Figures S7 and S8 illustrate the experimental data of $N_{down}/N_{up}$ under various operational conditions. The relationship between transmission efficiency and $Q_{out}$ is presented in Figure 8a, demonstrating a linear correlation with an increase in $Q_{out}$. The optimal ion transmission reached 54.3% at the maximum Qout of 3 L/min, paired with the lowest sheath flow velocity. In this setup, a significant negative voltage at the outlet electrode, combined with a grounded electrometer, generates an electric field that constantly draws THA$^+$ ions towards the DMA outlet. An adequate $Q_{out}$ is required to counterbalance the electrical velocity, as indicated by Eq (1). For particles with consistent mobility, a higher $V_{DMA}$ is essential to facilitate selection at increased sheath flow velocities. As sheath flow velocity escalates, ion transmission efficiency diminishes due to the inadequacy in compensating for electrical velocity. It is important to note that under the standard DMA P5 configuration, where both the outlet electrode and detectors are grounded, ion loss attributed to electrical drag is insignificant. Therefore, transmission efficiency becomes independent of $Q_{out}$. Hence, the findings presented here signify the lower threshold of ion transmission capability for the DMA P5. The ion transmission of the TSI 1 nm-DMA (Model 3086) was evaluated using the same TDMA system. Ion transmission rates were determined to be 4.7% and 7.1% for sheath/sample flow ratios of 25 L/min/2.5 L/min and 30 L/min/3.0 L/min, respectively. Compared to other cylindrical DMAs, the DMA P5 exhibited an ion transmission efficiency that was 4.5 to 17.5 times greater, as shown in Figure 8b.

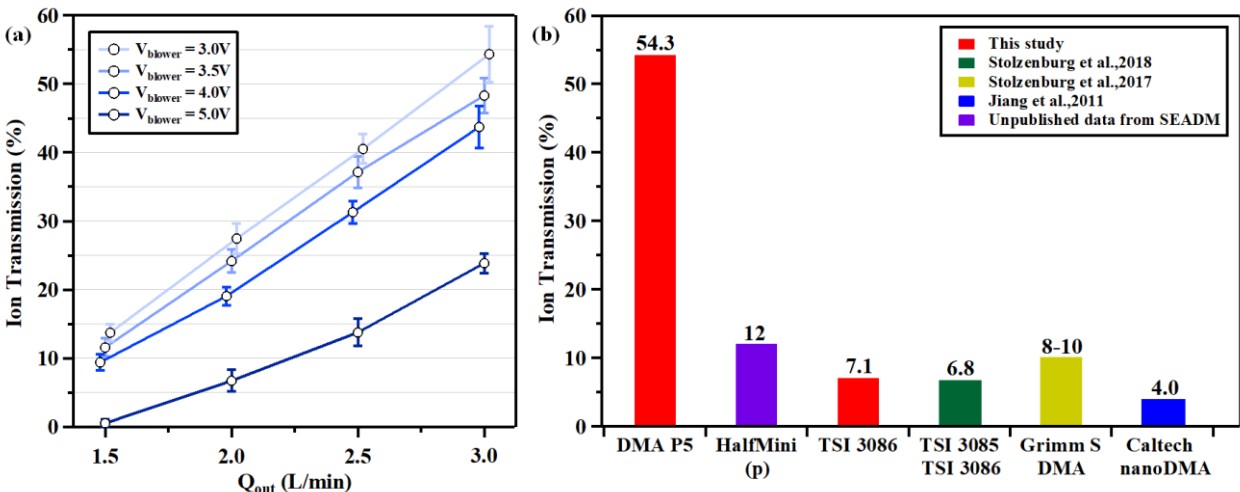

**Fig. 8 (a) Ion transmission efficiency of P5 under different $Q_{out}$; (b) Comparison with other cylindrical DMAs (the red bars represent the experimental results).**

### 3.3 Applications of DMA P5 by coupling with mass spectrometer

Commercial instruments that are capable of simultaneous ion mobility and mass measurements, such as ion mobility spectrometry-mass spectrometers (IMS-MS), are prevalent in the industry. Typically, these systems employ either intense electric fields within the mobility analyzer that may induce fragmentation of samples or conduct the analysis under conditions of reduced pressure (May and McLean, 2015). Unique to the DMA P5 is its operation at atmospheric pressure, which notably reduces the fragmentation of even weakly bound clusters, making it particularly apt for detecting atmospheric clusters. Sulfuric acid molecules and clusters, which play a pivotal role in new particle formation (Kulmala et al., 2000), were selected to validate the performance of the DMA P5 as an ion mobility filter for the examination of atmospheric relevant clusters. The experimental apparatus incorporated a nano-ESI ionization source connected to the DMA P5, with the latter output linked to a Lynx E11 electrometer and an API-TOF-MS (TOFWERK AG, Switzerland) via a customized interface (MION, Valladolid, Spain). The API-TOF-MS boasts an exceptionally low background noise level with a detection limit below 1 ion/cm³ (Junninen et al., 2010). Api-TOF and the combination of API-TOF with a chemical ionization source (CI-API-TOF) has been proven to be effective for the detection of various atmospheric clusters including sulfuric acid complexed with ammonia, amines, and organics in both laboratory and field investigations (Kirkby et al., 2011; Lehtipalo et al., 2016; Almeida et al., 2013; Yin et al., 2021; Riccobono et al., 2014). Coupling DMA P5 with API-TOF also facilitates the study of the physicochemical properties of atmospheric clusters by adjusting the electric field strength within the transfer optics (Lopez-Hilfiker et al., 2016). A solution containing 200 mM sulfuric acid in a 1:4 methanol/water mixture was employed to generate negatively charged sulfuric acid clusters via ESI, using the same procedure outlined in preceding sections. The mobility of these clusters was calibrated using THA⁺ ions before each experimental run. Hereafter, the nomenclature for the sulfuric acid clusters will be standardized as

follows: bisulfate ions ($HSO_4^-$), sulfuric acid dimers ($H_2SO_4HSO_4^-$), trimers (($H_2SO_4)_2HSO_4^-$), and tetramers (($H_2SO_4)_3HSO_4^-$) will be designated as sulfuric acid monomer $(SA)_1^-$, dimer $(SA)_2^-$, trimer $(SA)_3^-$, and tetramer $(SA)_4^-$, respectively.

**Table 1 mobilities 1/Z (V s$^{-1}$cm$^{-2}$) and mobility diameter (nm) of sulfuric acid clusters**

| Peak$^+$ | 1/Z | Diameter |
|---|---|---|
| $(SA)_1^-$ | 0.497 | 1.02 |
| $(SA)_2^-$ | 0.502 | 1.024 |
| $(SA)_3^-$ | 0.589 | 1.111 |
| $(SA)_4^-$ | 0.676 | 1.191 |

Figure 9 illustrates the ion mobility spectrum alongside the two-dimensional distribution of ion mobility and mass-to-charge ratio (DMA-MS spectrum). The ion mobility spectrum, depicted in Figure 9a, underwent multi-peak fitting analysis using the Igor Multi-peak Fitting package. The identified peaks corresponded with those in the DMA-MS spectrum, with distinctly resolved peaks for $(SA)_3^-$ and $(SA)_4^-$. However, the resolution was insufficient to separate $(SA)_1^-$ and $(SA)_2^-$ within the ion mobility spectrum. The ion mobility (diameter) for $(SA)_1^-$ and $(SA)_2^-$ was derived from the DMA-MS spectrum. Table 1

presents the ion mobility and diameter measurements for the sulfuric acid monomer and larger clusters. The measured diameter for $(SA)_3^-$ was 1.11 nm, aligning closely with the value reported by Passananti et al. (2019). It should be noted that the voltage settings can impact cluster fragmentation within the API-TOF-MS, making the DMA-MS spectrum highly contingent on the instrument configuration. Thus, caution is necessary when comparing results across different experiments. Considering that clusters which serve as precursors to atmospheric nucleation are often physically labile, a pivotal question emerges: are the

clusters detected in mobility and mass analyzers the actual atmospheric species, or are they fragmentation products created artificially during vacuum transfer? This complexity is magnified if fragmentation occurs before or during the ion mobility measurements. Consequently, factoring in fragmentation is vital for interpreting DMA-MS data. Ions with lower mass that exhibit identical mobility to parent ions typically result from dissociation or decomposition. Assuming not all parent ions fragment within the mass analyzer, it is possible to discern which detected ions are the original species selected by the DMA

and which are fragments. As depicted in Figure 9b, apart from SA multimers, a cluster of methyl sulfate with $(SA)_1^-$ ($CH_4SO_4HSO_4^-$) was detected at distinct mobilities ($V_{DMA}$ of approximately 1800 V and 2250 V). Additionally, a cluster of an ammonia molecule adducted to $(SA)_4^-$ was identified at a $V_{DMA}$ of around 2450V. The first three peaks in the mobility spectrum (Figure 9a) corresponded to $NO_2^-$, $CO_3^-$, and $CHO_4^-$. To further analyze fragmentation, the ion mobility spectrum at the mass-to-charge ratio of the primary observed ions/cluster-ion adducts was examined (Figure S9): both $(SA)_1^-$ and $CH_3SO_4^-$ exhibited

two peaks, with the latter identified as fragments derived from $CH_4SO_4HSO_4^-$. The mobilities of $(SA)_1^-$ and $(SA)_2^-$ were similar, complicating their separation; however, $(SA)_2^-$ fragmentation was unlikely to significantly contribute to $(SA)_1^-$, given the marked difference in their central mobilities. The subsequent peaks for $(SA)_2^-$ were fragments from $(SA)_3^-$ and the methyl sulfate cluster with $(SA)_2^-$ ($CH_4SO_4(H_2SO_4)HSO_4^-$). $(SA)_4^-$ was anticipated to fragment into $(SA)_3^-$ by losing a sulfuric acid

molecule, arising from the fragmentation of NH₃-(SA)₄⁻ and larger clusters. In our experimental setup, the fragmentation
impact on the intensity of SA clusters was minimal (<10%). When deploying the (CI)-API-TOF for atmospheric measurements,
it is crucial to verify the instrument configuration against a similar experimental setup to avoid underestimating cluster
concentration due to a high de-clustering ratio.

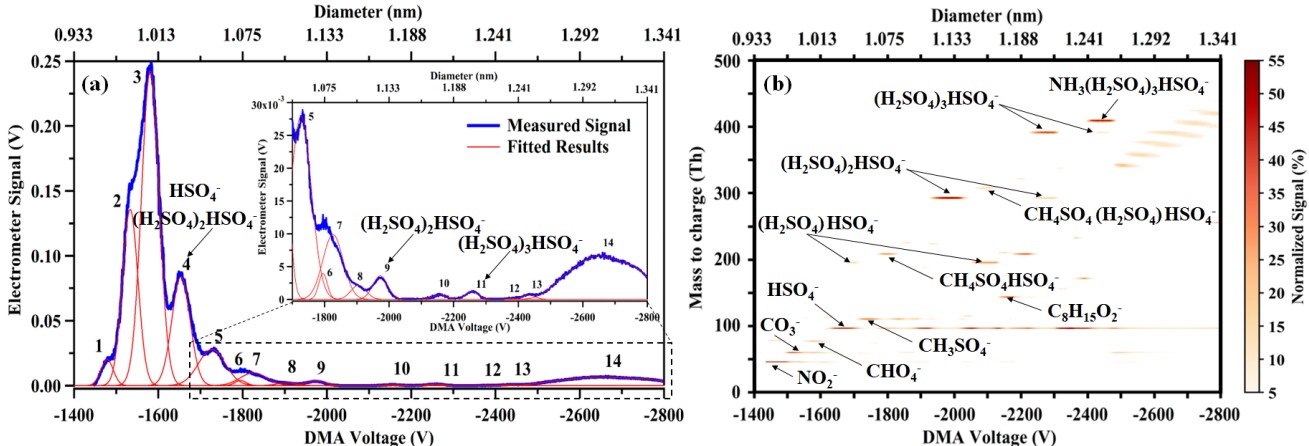

**Fig. 9 Negative ion mobility spectrum of electrosprayed sulfuric acid of (a) electrometer and (b) DMA P5 coupled to API-ToF-MS.**
**The signal of DMA-MS spectrum within each ion mobility step (V$_{DMA}$ increased with the stepwise of 0.5V) is normalized to the total**
**signal intensity of the corresponding ion mobility step.**

## 4 Conclusions and Recommendations

This study delineates the characterization outcomes for a planar differential mobility analyzer (DMA P5). The DMA P5
features a sizing scope below 3.9 nm and is operable in two distinct modes: suction and counter-flow. In suction mode, it
achieves a maximal resolving power of 60, while in counter-flow mode, it can attain up to 84. The resolving power of the
DMA P5 is demonstrated to be 5 to 16 times superior to that of conventional commercial DMAs. Within the suction mode,
the concentration of monodispersed aerosols can be tuned by adjusting the injection flow rate. Conversely, the counter-flow
mode, despite its enhanced resolving power, yields a lower concentration of monodispersed aerosols due to the lack of injection
flow. Ion transmission efficiency of the DMA P5, as assessed by a tandem differential mobility analyzer (TDMA) system,
surpasses 54.3%, which is approximately 7 to 8 times greater than that of the benchmark commercial DMA model (TSI 3086).
The utility of the DMA P5 was further explored by assessing both positive and negative aerosol ions of four tetra-alkyl
ammonium halides (THAB, TMAI, TBAI, and TDAB), from which high-resolution ion spectra were acquired. Subsequently,
the DMA P5 was integrated with an atmospheric pressure interface time-of-flight mass spectrometer (API-TOF-MS), enabling
the successful determination of the two-dimensional distribution of sulfuric acid clusters based on mass-to-charge ratio and
ion mobility. The mobility diameters of sulfuric acid clusters, ranging from monomer to tetramer, were precisely measured.

This integrated system affords the simultaneous measurement of ion mobility and chemical composition of atmospheric clusters. Moreover, it is applicable for calibrating the mass-dependent ion transmission efficiency of mass spectrometers and investigating the influence of collision-induced cluster fragmentation (CICF) within the mass spectrometer on atmospheric cluster measurements.

## Data Availability

Data available on request from the authors.

## Conflicts of Interest

The authors declare no conflict of interests.

## Author Contributions

**Zhengning Xu:** Methodology, System set up, Experiment, Formal analysis, Visualization, Writing Original draft, Review and editing. **Jian Gao, Zhuanghao Xu:** Experiment, System set up and Data curation. **Michel Attoui, Xiangyu Pei:** System set up, Review and editing. **Mario Amo-González, Kewei Zhang:** System set up. **Zhibin Wang:** Conceptualization, Project administration, Review and editing; Funding acquisition.

## Acknowledgments

The research was supported by National Natural Science Foundation of China (NSFC) (42005086, 41805100, 91844301), the Fundamental Research Funds for the Central Universities (226-2023-00077) and the Key Research and Development Program of Zhejiang Province (2021C03165 and 2022C03084).

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
