# Peer review of "Characterization of the planar differential mobility analyzer (DMA P5): resolving power, transmission efficiency and its application to atmospheric relevant cluster measurements"

_Atmospheric Measurement Techniques, 2023_

## Referee Comment (RC3)

Discussion of: **Characterization of the planar differential mobility analyzer (DMAP5): resolving power, transmission efficiency and its application to atmospheric cluster measurements**, by: Zhengning Xu, Jian Gao, Zhuanghao Xu, Michel Attoui, Xiangyu Pei, Mario Amo-González, Kewei Zhang, Zhibin Wang

by Juan Fernandez de la Mora, Yale University, Mechanical Engineering Department.

The article describes an instrument combination previously used in laboratory studies, but, to my knowledge, not in atmospheric studies. I see merit in that approach, and consider this exploratory article a valuable contribution to the field of atmospheric measurements. It covers prior work fairly, and adheres to high scientific standards. I support its publication, pending some desirable improvements. There is the important issue of whether the instrument will be sufficiently sensitive for relevant atmospheric studies. Whether or not this is the case, the article already shows clearly that the instrument can be very fruitful at least in laboratory studies of atmospherically relevant clusters. In this I believe there are precedents that ought to be cited.

**1. Abstract**. It would be preferable to avoid the abbreviations TMAI, TBAI, THAB and TDAB. Also the term "*newly developed*" seems inappropriate for an instrument that has existed for considerable time.

**2. DMA Transmission**. The transmission study is most valuable, as I am not aware of prior quantitative studies of this important metric for planar DMAs that would be directly relevant to atmospheric measurements. There are studies with electrospray sources directly facing a planar DMA inlet slit, showing many orders of magnitude gains in transmitted ion signal versus cylindrical DMAs. However, an electrospray is an intense unipolar ion source injecting several hundred nA of current into an area typically smaller than $1 \mu m^2$. In this case, the major source of ion loss is beam broadening by space charge, which would not be relevant in most atmospheric measurements. In a planar DMA the electrospraying capillary can be brought arbitrarily close to the inlet slit. Under such conditions, Javaheri et al. (2008) have shown that almost all molecules of an electrosprayed dilute species in solution can be transmitted as ions through the atmospheric inlet orifice of a mass spectrometer. Similarly, over 1 nA of the dominant electrosprayed ion can be transmitted to the outlet slit of a planar DMA (Fernandez de la Mora, 2019). This exceptional transmission has been exploited by Tauber et al. (2018) to introduce DMA-purified highly concentrated atomic ions into a nucleation chamber to study ion induced nucleation. Their study suggests that the same is not possible with any cylindrical DMA. Similarly, in our first tandem DMA-MS study involving a Time of Flight (TOF) mass spectrometer, it took us over an hour with a high resolution cylindrical DMA (passing 10 L/min of aerosol) set at a fixed voltage to accumulate a single mass spectrum with useful information on $(THABr)_n(THA^+)_z$ clusters (Fernandez de la Mora et al. 2005). This is the main reason why most successful couplings of DMAs with MS systems (mainly using electrospray sources) have involved planar DMAs. There are nevertheless exceptions involving quadrupole MS systems set at a fixed mass, while a cylindrical DMA scans over the voltage (Ude et al., 2004). Steiner et al. (2014) have successfully coupled a cylindrical DMA of relatively high resolution to a TOF-MS to investigate small ions from radioactive bipolar sources.

The present transmission study uses a tandem setup with a cylindrical first DMA. In this case the vast majority of electrospray ions are lost by space charge in the aerosol inlet tube and in the

annular region preceding the inlet slit to the analyzing region of the first (cylindrical) DMA. Consequently, there is little space charge left when these ions reach the second (planar) DMA. Therefore, the losses measured by Xu et al. are primarily diffusive losses, which is what would be relevant in atmospheric sampling. These losses are not negligible for the small ions investigated by Xu et al., and the observed substantial advantage of planar over cylindrical DMAs (Figure 5b) is apparently also due to lack of an extended annular region upstream of the injection slit. This dominant region of diffusive losses is evidently reduced in cylindrical DMAs using a small outer radius $R_2$, which explains the advantage of the Half-Mini DMA ($R_2$=7 mm) reported in figure 5b over other cylindrical DMAs.

It would be useful if the authors would report the geometry in the cubic chamber used in the planar DMA upstream of the injection slit, since this might be the major source of the observed 46% ion loss. Most relevant to these losses is how far from the slit is the end of the tube bringing in the aerosol into this cubic chamber.

**3. Mass spectrometer selection**. Xu et al. use a TOFWORK AG mass spectrometer. In this they follow the lead of many widely cited atmospheric studies by Kulmala and colleagues. However, most other past DMA-MS couplings have relied on other commercial time of flight mass spectrometers developed broadly for electrospray mass spectrometry studies, many of them having much higher resolving power and mass range than the TOFWORK MS. These other instruments have achieved high reliability and ion transmission efficiency, and would at first sight seem to be ideally fitted for coupling with a DMA for atmospheric studies. It would be of considerable interest to those intending to pursue related atmospheric DMA-MS studies to learn about the considerations that have led Xu et al. to their MS choice.

**4. Cluster fragmentation**

Given that the clusters forming as precursors to atmospheric nucleation are physically bound and often fairly labile, the important issue arises as to whether the clusters observed in the mobility and the mass analyzers are the original species present in the atmosphere, or rather their fragmentation products artificially generated during their relatively violent transfer to the vacuum system. The matter is briefly alluded to in line 270 "*Since the voltage configurations can affect the fragmentation of the cluster inside the API-ToF-MS (270 Passananti et al., 2019), the DMA-MS spectrum is highly instrument dependent*". Nevertheless, more discussion on fragmentation would be indicated in relation to an instrument put together to investigate atmospheric nucleation. Fragmentation is certainly strongly affected by the choice of the mass spectrometer and its voltage settings, and this may relate to point **3** above. There are excellent commercial instruments able to measure both mobility and mass in tandem. Most of them use either intense potentially fragmenting electric fields in the mobility analyzer, or carry the mobility analysis in a region of reduced pressure. What is special about the DMA is that it operates at atmospheric pressure and has little tendency to fragment even weakly bound clusters. It is accordingly possible to establish which ions detected in the MS are the original ions selected in the DMA, and which are fragments. This possibility is much more limited in situations when fragmentation may arise prior to or during the mobility measurement. This important advantage of the DMA is well illustrated in the work cited by Hogan and colleagues. It is also nicely demonstrated in the rather interesting DMA-MS spectrum included in Figure 9b of Xu et al. This lovely figure seems to me to deserve far more discussion than currently provided. For instance, if the bisulfate dimer ion had fragmented into a monomer during its transit to the vacuum system, an ion with the mobility of the dimer and the mass of the monomer would appear in Figure 9b. It is not clear in that figure if this fragmentation product is present or not, but

the same deconvolution used in Figure 9a would clarify the issue. On the other hand, it is certain from Figure 9b that the bisulfate trimer does not decompose into either a dimer or a monomer. Yet the tetramer does undoubtedly decompose partially into the trimer during its vacuum transit. This new experimental tool is therefore already telling us a lot about how the stability of these clusters changes with their size. It would be most helpful if the authors would refer to prior literature on the stability of sulfate or bisulfate clusters.

There are a number of other transitions revealed by figure 9b, whose less obvious potential relevance would call for additional input from the authors. For instance, a mass a little larger than that of the dimer (perhaps a solvated dimer) arises at voltages of about 1800 and 2250 Volts. These two ions decompose partly into the monomer on their way to the MS, providing some additional basis to the guess that they are indeed solvated bisulfate dimers. Similarly, an ion slightly heavier than the tetramer (perhaps a solvated tetramer) decomposes into the tetramer.

Another potentially interesting feature in Figure 9a is the presence of an ion at approximately half the monomer mass. Please, clarify if this is the doubly charged sulfate.

**5. Minor remarks**
*Following equation (1), $U$ should rather be the velocity in the symmetry plane going through the center of the slit.

*The article states that "*The reason for the difference of resolving power between the two recirculation modes and the deviation from the theoretical calculation is the turbulence effect.*" I have my doubts about this interpretation. What would be its basis?

*Line 244 states "*the resolving power of planar DMA is directly related to $V_{DMA}$ and Ne.*" Does $Ne$ refer to the negative spray? Please clarify the relevance of this, as it is not at all clear.

*Line 165: The program *Igor* is quoted for mobility peak analysis. Would you please provide a little more background for those unfamiliar with this tool?

* The authors note that their recirculation circuit is not part of the commercial system, perhaps to warn readers of the possibility that DMA performance may depend on this component of the system. I doubt that the flow control part will be much effect on DMA performance, though I may be wrong. One original component in this recirculating flow system perhaps deserves some comment. This is the planar commercial HEPA filter, apparently sandwiched between two surfaces with NW-40 connectors. Would the authors please provide some more detail of this design?

* The reference to Fernandez de la Mora and Kozlowski given in Figure 5b must be incorrect, as their study did not include transmission measurements. The correct reference must be a later study by Attoui and colleagues.

**6. Conflict of interest statement.**
JFM collaborates frequently with authors Michel Attoui and JFM's former graduate student Mario Amo-Gonzalez.

JFM and his wife owned half of the now bankrupt company SEADM where Mario Amo Gonzalez led the development of the planar DMA P5. JFM remains keenly interested in the continuation of SEADM's efforts by others, including the company MION SL.

**References**

Fernandez de la Mora, J. (2019) Space charge effects in ion mobility spectrometry, J. American Society Mass Spectrometry, 30(6), 1082-1091

Fernandez de la Mora, J.; Bruce Thomson and M. Gamero-Castaño (2005), Tandem mobility mass spectrometry study of electrosprayed Heptyl$_4$N$^+$Br$^-$ clusters, J. Am. Soc. Mass Spectrom., 16 (5): 717-732.

Javaheri, H., Le Blanc, Y., Thomson, B. A., de la Fernandez Mora, J., Rus, J., Sillero-Sepúlveda, J.A. (2008) Evaluation of the analytical characteristic of a differential mobility analyzer coupled to a triple quadrupole system (DMA-MSMS), Poster 061, Annual meeting of the ASMS, June 1-5, Denver, CO.

Steiner, G; Jokinen, T; Junninen, H; Sipila, M; Petaja, T; Worsnop, D; Reischl,; Kulmala, M. (2014) High-Resolution Mobility and Mass Spectrometry of Negative Ions Produced in a Am-241 Aerosol Charger, Aerosol Sci. & Techn. 48(3) 261-270 DMA-TOF

Tauber, C.; X. Chen, P.E. Wagner, P.M. Winkler, C.J. Hogan Jr., A. Maißer (2018) Heterogeneous nucleation onto monoatomic ions: support for the Kelvin-Thomson theory, ChemPhysChem 19 3144–3149.

Ude, S; J. Fernandez de la Mora, B. A. Thomson, Charge-induced unfolding of multiply charged polyethylene glycol ions, J. Am. Chem. Soc., **126**, 12184-12190 (2004).

---

## Author Comment (AC3)

**Referee #3**

Discussion of: **Characterization of the planar differential mobility analyzer (DMA P5): resolving power, transmission efficiency and its application to atmospheric cluster measurements**, by: Zhengning Xu, Jian Gao, Zhuanghao Xu, Michel Attoui, Xiangyu Pei, Mario Amo-González, Kewei Zhang, Zhibin Wang

by Juan Fernandez de la Mora, Yale University, Mechanical Engineering Department.

The article describes an instrument combination previously used in laboratory studies, but, to my knowledge, not in atmospheric studies. I see merit in that approach, and consider this exploratory article a valuable contribution to the field of atmospheric measurements. It covers prior work fairly, and adheres to high scientific standards. I support its publication, pending some desirable improvements. There is the important issue of whether the instrument will be sufficiently sensitive for relevant atmospheric studies. Whether or not this is the case, the article already shows clearly that the instrument can be very fruitful at least in laboratory studies of atmospherically relevant clusters. In this I believe there are precedents that ought to be cited.

**1.Abstract.**

It would be preferable to avoid the abbreviations TMAI, TBAI, THAB and TDAB. Also the term "newly developed" seems inappropriate for an instrument that has existed for considerable time.

We thank the reviewer for the comment. We have removed the abbreviations in the abstract in the revised manuscript. We agreed with the reviewer that it is inappropriate to described DMA P5 as newly developed, since Amo-González et al. have coupled DMA P5 with mass spectrometer in 2018. We have removed the "*newly developed*" in the revised manuscript.

*Reference*

*Amo-González, M. and Pérez S.: Planar Differential Mobility Analyzer with a Resolving Power of 110, Analytical Chemistry, 90, 6735–6741, 10.1021/acs.analchem.8b00579, 2018*

**2.DMA Transmission.**

The transmission study is most valuable, as I am not aware of prior quantitative studies of this

important metric for planar DMAs that would be directly relevant to atmospheric measurements. There are studies with electrospray sources directly facing a planar DMA inlet slit, showing many orders of magnitude gains in transmitted ion signal versus cylindrical DMAs. However, an electrospray is an intense unipolar ion source injecting several hundred nA of current into an area typically smaller than $1\mu m^2$. In this case, the major source of ion loss is beam broadening by space charge, which would not be relevant in most atmospheric measurements. In a planar DMA the electrospraying capillary can be brought arbitrarily close to the inlet slit. Under such conditions, Javaheri et al. (2008) have shown that almost all molecules of an electrosprayed dilute species in solution can be transmitted as ions through the atmospheric inlet orifice of a mass spectrometer. Similarly, over 1 nA of the dominant electrosprayed ion can be transmitted to the outlet slit of a planar DMA (Fernandez de la Mora, 2019). This exceptional transmission has been exploited by Tauber et al. (2018) to introduce DMA-purified highly concentrated atomic ions into a nucleation chamber to study ion induced nucleation. Their study suggests that the same is not possible with any cylindrical DMA. Similarly, in our first tandem DMA-MS study involving a Time of Flight (TOF) mass spectrometer, it took us over an hour with a high resolution cylindrical DMA (passing 10 L/min of aerosol) set at a fixed voltage to accumulate a single mass spectrum with useful information on (THABr)n(THA+)z clusters (Fernandez de la Mora et al. 2005). This is the main reason why most successful couplings of DMAs with MS systems (mainly using electrospray sources) have involved planar DMAs. There are nevertheless exceptions involving quadrupole MS systems set at a fixed mass, while a cylindrical DMA scans over the voltage (Ude et al., 2004). Steiner et al. (2014) have successfully coupled a cylindrical DMA of relatively high resolution to a TOF-MS to investigate small ions from radioactive bipolar sources.

The present transmission study uses a tandem setup with a cylindrical first DMA. In this case the vast majority of electrospray ions are lost by space charge in the aerosol inlet tube and in the annular region preceding the inlet slit to the analyzing region of the first (cylindrical) DMA. Consequently, there is little space charge left when these ions reach the second (planar) DMA. Therefore, the losses measured by Xu et al. are primarily diffusive losses, which is what would be relevant in atmospheric sampling. These losses are not negligible for the small ions investigated by Xu et al., and the observed substantial advantage of planar over cylindrical DMAs (Figure 5b) is apparently also due to lack of an extended annular region upstream of the injection slit. This dominant region of diffusive

losses is evidently reduced in cylindrical DMAs using a small outer radius R2, which explains the advantage of the Half-Mini DMA (R2=7 mm) reported in figure 5b over other cylindrical DMAs. It would be useful if the authors would report the geometry in the cubic chamber used in the planar DMA upstream of the injection slit, since this might be the major source of the observed 46% ion loss. Most relevant to these losses is how far from the slit is the end of the tube bringing in the aerosol into this cubic chamber.

We thank the reviewer for the comment. The reviewer has listed numbers of outstanding works from the precedents about the ion transmission study of planar DMA and the comparison with cylindrical DMAs. Moreover, the reviewer also well illustrated the benefit of applying TDMA on studying the ion transmission of DMA P5 for atmospheric relevant study. We have cited these works in the revised manuscript.

Moreover, as being pointed out, the geometry of cubic chamber is related to the reported ion transmission efficiency. The inner geometry of cubic chamber was cylinder-shaped, with the diameter of 32 mm, and the height of 6.5 mm. A straight conductive silicone tube is inserted into the center of the cylinder-shaped space thorough the hole on the top of the chamber. The space between the conductive silicone tube and the hole is sealed with silicone glue. The injection end of the tube is connected to the beam splitter, while the exit end is at the same plane with the lower end of the chamber. When the chamber is sealed to the inlet electrode of DMA P5, the distance between the exit end of tube and the inlet slit of DMA P5 is about 5.2 mm. In the revised manuscript, the geometry of cubic chamber was reported. The detailed description of how the monodispersed $THA^+$ ions are injected to the inlet slit was also added in the revised manuscript. The added sentences are shown as following "….. *Downstream to the Half Mini DMA, DMA P5 operated at scan mode was connected between the flow splitter and the second electrometer. A 3D-printed cubic chamber was used was used to connect the inlet tube and DMA inlet slit. The inner geometry of cubic chamber was cylinder-shaped, with the diameter of 32 mm, and the height of 6.5 mm. The inlet tube was inserted straightly into the centre of the cylinder-shaped space thorough the hole on the top of the cubic chamber. The space between the conductive silicone tube and the hole was sealed with silicone glue. The injection end of the inlet tube was connected to the beam splitter, while the exit end was at the same surface with the lower end of the chamber. When the cubic chamber was sealed to the inlet electrode of DMA P5, the distance between the exit end of inlet tube and*

*the inlet slit of DMA P5 is about 5.2 mm. …..".* We hope other group using DMA P5 can find useful information from our work.

**3. Mass spectrometer selection**

Xu et al. use a TOFWORK AG mass spectrometer. In this they follow the lead of many widely cited atmospheric studies by Kulmala and colleagues. However, most other past DMA-MS couplings have relied on other commercial time of flight mass spectrometers developed broadly for electrospray mass spectrometry studies, many of them having much higher resolving power and mass range than the TOFWORK MS. These other instruments have achieved high reliability and ion transmission efficiency, and would at first sight seem to be ideally fitted for coupling with a DMA for atmospheric studies. It would be of considerable interest to those intending to pursue related atmospheric DMA-MS studies to learn about the considerations that have led Xu et al. to their MS choice.

We thank the reviewer for the comment. There were two main reasons that we chose the APi-TOF from TOFWORK AG to couple with DMA P5 for atmospheric relevant clusters study: *(1) Low detection limit*: the APi-TOF consists of a time-of-flight mass spectrometer (TOF) coupled to an atmospheric pressure interface (APi). The Api has three differentially pumped chambers, the first two containing short segmented quadrupoles used in ion guide mode, and the third containing an ion lens assembly. The APi-TOF has a very low background noise level and detection limit. Junninen et al., (2010) reported that the detection limit of the APi-TOF within the mass range of 80-900 was below 1 ion/cm$^3$. Taking the high ion transmission of DMA P5 into consideration, we think coupling DMA P5 with the APi-TOF have the potential for the measurement of atmospheric clusters. Moreover, both Api-TOF and Api-TOF coupling with a chemical ionization source (CI-API-TOF) have been successfully applied for atmospheric cluster measurement in laboratory and field, exhibiting the capability of detecting different atmospheric clusters, such as cluster complexes of sulfuric acid with ammonia (Kirkby et al., 2011; Lehtipalo et al., 2016), amine (Almeida et al., 2013; Yin et al., 2021), and organics (Riccobono et al., 2014). *(2)* **Adjustable APi configuration:** Fragmentation of molecular clusters inside MS is a significant source of uncertainty in a wide range of chemical applications. Different clusters fragmented differently inside MS due to the different binding energy (*Lopez-Hilfiker et al., 2016; Passananti et al., 2019*). The voltage configuration, combination of the voltages applied to the APi-TOF, can significantly affects the transmission and

fragmentation of clusters. By the combination of API-TOF and DMA-P5, we aim to study the physiochemical properties of the selected atmospheric (relevant) clusters (based on their ion mobility) by adjusting the voltage configuration, scanning the electric field strength within the transfer optics in real time while measuring a steady-state distribution of the selected clusters. With this function, we can experimentally determine the electric field strength required to break apart atmospheric (relevant) clusters, which are directly related to the binding energy of the clusters. The related experimental results are not within the scope of this work, and will be reported in another paper, currently under preparation. In the revised manuscript, we have added the consideration of why we choose API-TOF to couple with DMA P5. The revised sentences are shown as following

"……. *API-ToF-MS has a very low background noise level and detection limit (< 1 ion/cm3, Junninen et al., 2010). Moreover, both Api-TOF and Api-TOF coupling with a chemical ionization source (CI-API-TOF) have been successfully applied for atmospheric cluster measurement in laboratory and field, exhibiting the capability of detecting different atmospheric clusters, such as cluster complexes of sulfuric acid with ammonia (Kirkby et al., 2011; Lehtipalo et al., 2016), amine (Almeida et al., 2013; Yin et al., 2021), and organics (Riccobono et al., 2014). By the combination of API-TOF and DMA-P5, it is also possible to study the physiochemical properties of the atmospheric relevant clusters scanning the electric field strength within the transfer optics (Lopez-Hilfiker et al., 2016). ……..*".

*Reference*

*Passananti, M., Zapadinsky, E., Zanca, T., Kangasluoma, J., Myllys, N., Rissanen, M. P., Kurtén, T., Ehn, M., Attoui, M., Vehkamäki, H., How well can we predict cluster fragmentation inside a mass spectrometer? Chemical Communications 2019, 55 (42), 5946-5949.*

*Kirkby, J., Curtius, J., Almeida, J., Dunne, E., Duplissy, J., Ehrhart, S., et al. (2011). Role of sulphuric acid, ammonia and galactic cosmic rays in atmospheric aerosol nucleation. Nature 476 (7361), 429–433. doi:10.1038/nature10343.*

*Lehtipalo, K., Rondo, L., Kontkanen, J., Schobesberger, S., Jokinen, T., Sarnela, N., et al. (2016). The effect of acid-base clustering and ions on the growth of atmospheric nano-particles. Nat. Commun. 7, 11594. doi:10.1038/ncomms11594.*

*Almeida, J., Schobesberger, S., Kurten, A., Ortega, I. K., Kupiainen-Maatta, O., Praplan, A. P., et al. (2013). Molecular understanding of sulphuric acid-amine particle nucleation in the atmosphere.*

*Nature 502 (7471), 359–363. doi:10.1038/nature12663.*

*Yin, R., Yan, C., Cai, R., Li, X., Shen, J., Lu, Y., et al. (2021). Acid-base clusters during atmospheric new particle formation in urban beijing. Beijing: Environmental Science & Technology. doi:10.1021/acs.est.1c02701.*

*Riccobono, F., Schobesberger, S., Scott, C. E., Dommen, J., Ortega, I. K., Rondo, L., et al. (2014). Oxidation products of biogenic emissions contribute to nucleation of atmospheric particles. Science 344 (6185), 717–721. doi:10.1126/science.1243527.*

*Junninen, H., Ehn, M., Petaja, T., Luosuarvi, L., Kotiaho, T., Kostiainen, R., Rohner, U., Gonin, M., Fuhrer, K., Kulmala, M., and Worsnop, D. R.: A high-resolution mass spectrometer to measure atmospheric ion composition, Atmos. Meas. Tech., 3, 1039–1053, doi:10.5194/amt-3-1039-2010, 2010.*

*Lopez-Hilfiker, F. D., Iyer, S., Mohr, C., Lee, B. H., D'Ambro, E. L., Kurtén, T., and Thornton, J. A.: Constraining the sensitivity of iodide adduct chemical ionization mass spectrometry to multifunctional organic molecules using the collision limit and thermodynamic stability of iodide ion adducts, Atmos. Meas. Tech., 9, 1505–1512, https://doi.org/10.5194/amt-9-1505-2016, 2016.*

**4. Cluster fragmentation**

Given that the clusters forming as precursors to atmospheric nucleation are physically bound and often fairly labile, the important issue arises as to whether the clusters observed in the mobility and the mass analyzers are the original species present in the atmosphere, or rather their fragmentation products artificially generated during their relatively violent transfer to the vacuum system. The matter is briefly alluded to in line 270 "Since the voltage configurations can affect the fragmentation of the cluster inside the API-ToF-MS (270 Passananti et al., 2019), the DMA-MS spectrum is highly instrument dependent". Nevertheless, more discussion on fragmentation would be indicated in relation to an instrument put together to investigate atmospheric nucleation. Fragmentation is certainly strongly affected by the choice of the mass spectrometer and its voltage settings, and this may relate to point 3 above. There are excellent commercial instruments able to measure both mobility and mass in tandem. Most of them use either intense potentially fragmenting electric fields in the mobility analyzer, or carry the mobility analysis in a region of reduced pressure. What is special about the DMA is that it operates at atmospheric pressure and has little tendency to fragment even weakly bound clusters. It is accordingly possible to establish which ions detected in the MS

are the original ions selected in the DMA, and which are fragments. This possibility is much more limited in situations when fragmentation may arise prior to or during the mobility measurement. This important advantage of the DMA is well illustrated in the work cited by Hogan and colleagues. It is also nicely demonstrated in the rather interesting DMA-MS spectrum included in Figure 9b of Xu et al. This lovely figure seems to me to deserve far more discussion than currently provided. For instance, if the bisulfate dimer ion had fragmented into a monomer during its transit to the vacuum system, an ion with the mobility of the dimer and the mass of the monomer would appear in Figure 9b. It is not clear in that figure if this fragmentation product is present or not, but the same deconvolution used in Figure 9a would clarify the issue. On the other hand, it is certain from Figure 9b that the bisulfate trimer does not decompose into either a dimer or a monomer. Yet the tetramer does undoubtedly decompose partially into the trimer during its vacuum transit. This new experimental tool is therefore already telling us a lot about how the stability of these clusters changes with their size. It would be most helpful if the authors would refer to prior literature on the stability of sulfate or bisulfate clusters. There are a number of other transitions revealed by figure 9b, whose less obvious potential relevance would call for additional input from the authors. For instance, a mass a little larger than that of the dimer (perhaps a solvated dimer) arises at voltages of about 1800 and 2250 Volts. These two ions decompose partly into the monomer on their way to the MS, providing some additional basis to the guess that they are indeed solvated bisulfate dimers. Similarly, an ion slightly heavier than the tetramer (perhaps a solvated tetramer) decomposes into the tetramer. Another potentially interesting feature in Figure 9a is the presence of an ion at approximately half the monomer mass. Please, clarify if this is the doubly charged sulfate.

We thank the reviewer for the comment. We have added more discussion on the cluster fragmentation. The revised manuscript is shown as following "……. *Since the voltage configurations can affect the fragmentation of the cluster inside the API-ToF-MS (Passananti et al., 2019), the DMA-MS spectrum is highly instrument dependent. Cautions should be paid on the comparison between different experiments. Given that the clusters forming as precursors to atmospheric nucleation are physically bound and often labile, the important issue arises as to whether the clusters observed in the mobility and the mass analyzers are the original species present in the atmosphere, or rather their fragmentation products artificially generated during the transfer to the vacuum system. The situations are even more complicated if fragmentation*

*arise prior to or during the mobility measurements. Therefore, it is important to consider the fragmentation when interpreting the DMA-MS measurements. Ions with smaller mass but appearing at the same mobility of parent ions are originated from dissociation or decomposition. Under the condition that not all parent ions are fragmented into smaller ions within the mass analyzer, we can determine which ions detected in the MS are the original ions selected in the DMA, and which are fragments. As shown in Fig. 9b, except for SA multimers, cluster of methyl sulfate with $(SA)_1^-$ $(CH_4SO_4HSO_4^-)$ was observed at the different mobility (VDMA of about 1800 V and 2250 V). Cluster of ammonia molecule adducted on $(SA)_4^-$ was also identified at VDMA of about 2450V. The first three peaks identified in the mobility spectrum (Fig 9a) were $NO_2^-$, $CO_3^-$ and $CHO_4^-$. To further interpret the fragmentation, the ion mobility spectrum at the mass to charge ratio of the main observed ions/cluster-ion adducts was discussed (Fig S6): both $(SA)_1^-$ and $CH_3SO_4^-$ showed two peaks, with the latter one being fragments originated from $CH_4SO_4HSO_4^-$. The mobility of $(SA)_1^-$ and $(SA)_2^-$ was close, making the separation of $(SA)_1^-$ and $(SA)_2^-$ fragments difficult. Hower, it is unlikely that $(SA)_2^-$ fragmented contributed largely to $(SA)_1^-$, due to an obvious difference in centroid mobility. The latte two peaks of $(SA)_2^-$ were the fragments from $(SA)_3^-$ and cluster of methyl sulfate with $(SA)_2^-$ $(CH_4SO_4 (H_2SO_4) HSO_4^-)$. $(SA)_4^-$ would fragment into $(SA)_3^-$ via losing one sulfuric acid molecule and originate from the fragmentation of $NH_3$-$(SA)_4^-$ and larger clusters. Under our experimental configuration, the interference of fragmentation on the intensity of SA clusters is minor (<10%). When the (CI)-API-TOF was deployed for atmospheric measurements, the instrument configuration should be checked by using similar experimental set up to avoiding underestimate of the clusters due to large de-cluster ratio.*".

**5. Minor remarks**

*Following equation (1), U should rather be the velocity in the symmetry plane going through the center of the slit.

We thank the reviewer for the comment. We have corrected the description of U in the revised manuscript.

*The article states that "The reason for the difference of resolving power between the two recirculation modes and the deviation from the theoretical calculation is the turbulence effect." I

have my doubts about this interpretation. What would be its basis?

We thank the reviewer for the comment. Firstly, we used the corrected formula to calculated the resolving power. The new formula is corrected from Amo-González et al. (2018). The derivations are added in the revised SI. The current resolving power, similar with the pre-delivery test at SEADM, is lower than the theoretical values. As shown in Eq. (4), except for the flow rates of mono/ploy-dispersed aerosol, and the geometry of DMA P5, the term affects the resolving power is the Reynolds number. We have compared with the reported experimental results with Amo-González et al. (2018), our results were approximately the same with the reported results (R=79, under $V_{DMA}$= 5500, for THA$^+$ monomer) without adding prelaminarizers into the recirculation system, which supports the Eq. (4). Moreover, the main difference between the two recirculation modes is that under suction mode polydiserpersed aesosol (at the flow rate of 2L/min) was injected from ESI chamber into the separation region from the inlet slit, while under counter flow mode countflow (at the flow rate of 0.5-1L/min) flow out of the separation region through the inlet slit. We think the way the ions inserted, which leads to different laminar condition of the sheath flow, results in the difference of the resolving power. As the decreased laminar condition is likely due to the extra turbulence effect for the flow injection. Consequently, we made the statement that it is the laminar condition that lead to the lower resolving power.

*References*

*Amo-González, M. and Pérez S.: Planar Differential Mobility Analyzer with a Resolving Power of 110, Analytical Chemistry, 90, 6735–6741, 10.1021/acs.analchem.8b00579, 2018*

*Line 244 states "the resolving power of planar DMA is directly related to VDMA and Ne." Does Ne refer to the negative spray? Please clarify the relevance of this, as it is not at all clear.

We thank the reviewer for the comment. Ne represents the net charge of the aerosol. We have added the description of Ne in the revised manuscript.

*Line 165: The program Igor is quoted for mobility peak analysis. Would you please provide a little more background for those unfamiliar with this tool?

We thank the reviewer for the comment. We have added the description of the analysis package used for multipeak fitting. We used Multi-peak Fitting 2 package embedded in Igor program for the peak fitting. This package was designed to provide curve fits to multiple, overlapping peaks of any sort

of measurement that results in localized peaks or lines. The program can also give you an estimate of the peak heights, widths, locations and areas.

* The authors note that their recirculation circuit is not part of the commercial system, perhaps to warn readers of the possibility that DMA performance may depend on this component of the system. I doubt that the flow control part will be much effect on DMA performance, though I may be wrong. One original component in this recirculating flow system perhaps deserves some comment. This is the planar commercial HEPA filter, apparently sandwiched between two surfaces with NW-40 connectors. Would the authors please provide some more detail of this design? *

We thank the reviewer for the comment. The original intension of mention the home-build recirculation system is that all components can be purchased from the local market. Our reported transmission and resolution are expected to be reproduced under suction mode or counter flow mode of DMA P5 with any recirculation system providing temperature-constant, aerosol-free, stable sheath flow. We agree with the reviewer that emphasizing home-build recirculation system can lead to the misunderstanding that the reported results may not be able to apply to other P5 DMA systems due to the difference in the recirculation system, which is not our original intention. In the revised manuscript, we have removed the descriptions of "***the home-build***", emphasizing the universal characteristics of the recirculation system that should be applied within DMA P5 system in section.

We have provided more details of the design how we put the HEPA filter into the recirculation in the revised manuscript. The provided description is shown as following "***The particle filter consists a planar commercial HEPA filter (Ref 34230010, Megalem MD143P3, Camfil Farr) and two stainless assembly. Top side of the assembly is a NW40 connector, while the bottom side fits the geometry of the planar HEPA filter. The HEPA filter is sandwiched between the two bottom surfaces of the two assembly, sealed with O-ring and screws.***"

The reference to Fernandez de la Mora and Kozlowski given in Figure 5b must be incorrect, as their study did not include transmission measurements. The correct reference must be a later study by Attoui and colleagues.

We thank the reviewer for the comment. We have corrected the reference in the revised manuscript. The reported transmission efficiency for HalfMini (p) is the internal data from SEADM.

**6. Conflict of interest statement.**

JFM collaborates frequently with authors Michel Attoui and JFM's former graduate student Mario Amo-Gonzalez. JFM and his wife owned half of the now bankrupt company SEADM where Mario Amo Gonzalez led the development of the planar DMA P5. JFM remains keenly interested in the continuation of SEADM's efforts by others, including the company MION SL.

**References**

Fernandez de la Mora, J. (2019) Space charge effects in ion mobility spectrometry, J. American Society Mass Spectrometry, 30(6), 1082-1091

Fernandez de la Mora, J.; Bruce Thomson and M. Gamero-Castaño (2005), Tandem mobility mass spectrometry study of electrosprayed Heptyl4N+Br- clusters, J. Am. Soc. Mass Spectrom., 16 (5): 717-732.

Javaheri, H., Le Blanc, Y., Thomson, B. A., de la Fernandez Mora, J., Rus, J., Sillero-Sepúlveda, J.A. (2008) Evaluation of the analytical characteristic of a differential mobility analyzer coupled to a triple quadrupole system (DMA-MSMS), Poster 061, Annual meeting of the ASMS, June 1-5, Denver, CO.

Steiner, G; Jokinen, T; Junninen, H; Sipila, M; Petaja, T; Worsnop, D; Reischl,; Kulmala, M. (2014) High-Resolution Mobility and Mass Spectrometry of Negative Ions Produced in a Am241 Aerosol Charger, Aerosol Sci. & Techn. 48(3) 261-270 DMA-TOF

Tauber, C.; X. Chen, P.E. Wagner, P.M. Winkler, C.J. Hogan Jr., A. Maißer (2018) Heterogeneous nucleation onto monoatomic ions: support for the Kelvin-Thomson theory, ChemPhysChem 19 3144–3149.

Ude, S; J. Fernandez de la Mora, B. A. Thomson, Charge-induced unfolding of multiply charged polyethylene glycol ions, J. Am. Chem. Soc., 126, 12184-12190 (2004).

---

## Author Response (AR1)

**Referee #1**

This manuscript presents the characterization of a high-resolution planar DMA for cluster classification. I recommend it to be published in Atmospheric Measurement Techniques considering its topic, as transmission and resolution of a DMA are important to the quantification of the measured clusters and nanoparticles. However, I am confused by the presentation in several aspects and recommend a major revision to the manuscript.

Major comments:

1. A major concern is how the audience digest the results and use them in their future studies. The authors need to clarify whether the transmission and resolution are specific to their experimental setup or whether other P5 DMAs are expected to share similar values. Detailed comments are given below:

a. A home-build recirculation system is mentioned several times, making the manuscript very confusing. A home-build system may indicate that the results of this study do not apply to other P5 DMA systems due to the difference in the recirculation system. More discussions are needed when reporting the values from this system: do the authors expect similar transmission and resolution in other P5 systems as a function of different working conditions?

We thank the reviewer for this comment. The original intension of the mention of the home-build recirculation system is that all components can be purchased from the local market. Our reported transmission and resolution are expected to be reproduced under suction mode or counter flow mode of DMA P5 with any recirculation system providing temperature-constant, aerosol-free, stable sheath flow. We agree with the reviewer that emphasizing home-build recirculation system can lead to the misunderstanding that the reported results may not be able to apply to other P5 DMA systems due to the difference in the recirculation system, which is not our original intention.

In the revised manuscript, we have removed the descriptions of "***the home-build***", emphasizing the universal characteristics of the recirculation system that should be applied within DMA P5 system in Section 2.

The revised sentences are shown as following: "***The recirculation circuit needed for DMA P5 system should be able to provide particle free sheath flow with stable velocity and temperature. The recirculation circuit deployed in this study consists of air blower (Ref 497.3.265-361, Domel),***

*water cooler coupled with constant temperature water bath (DCW-2008, SCIENTZ), particle filter adapted for high flow velocity, NW40 and NW 50 corrugated stainless-steel tubes and connectors. The particle filter consists a planar commercial HEPA filter (Ref 34230010, Megalem MD143P3, Camfil Farr) and two stainless assembly. The top side of the assembly is a NW40 connector, while the bottom side fits the geometry of the planar HEPA filter. The HEPA filter is sandwiched between the two bottom surfaces of the assembly, sealed with O-ring and screws. All the components are purchased from the local market. Alternative products with similar performance should not affect the operation of the whole system.*" (Line 71-80)

b. More figures and tables on the performance of the P5 DMA under different working conditions will be appreciated. The aim is to help the audience to estimate the resolution and transmission of the P5 DMA in their studies without repeating the same calibration experiment. These figures and tables can go to the SI.

We thank the reviewer for this comment. We have added more figures about the obtained mobility spectrum of standard ions under different working condition and the experimental results of transmission characterization into the SI.

The added Figure is Figure S3, Figure S7 and Figure S8, which are shown below:

[Figure]

**Figure S3 The positive ion mobility spectrum of THAB under suction mode ($V_{blower}$ = 5V, $Q_{in}$ = 5L/min, $Q_{out}$= 1.5L/min) with different solution concentrations**

[Figure]

**Figure S7 The distribution of transmission efficiency of the DMA P5 when classifying THA[+] _under different sheath flow rate, with $Q_{out}$= 2.5 L/min**

[Figure]

**Figure S8 The distribution of transmission efficiency of the DMA P5 when classifying THA[+] _under different sheath flow rate, with $Q_{out}$= 3.0 L/min**

c. Transmission vs penetration. The use of the transmission is rather confusing. I spent quite some time checking section 3.2 and found the authors made no mistakes. However, I am afraid that some readers may be misled as penetration is 1 - particle loss, while transmission is also affected by the peak shape. Their relationship is

transmission = penetration * peakHeight of transfer Function without Loss

For an ideal DMA for sub-micron particles at balanced flow rates, peakHeight of transfer Function without tLoss = 1. However, for clusters and nanoparticles, this value is less than 1.0 as diffusion, turbulence, and other non-idealness will decrease the peak height even though there is no particle loss. Using the transmission may be straightforward when converting the cluster signal to concentration in DMA-MS measurement with a known signal-component cluster sample, yet penetration is used when inverting the signal to cluster/aerosol size distribution. More importantly, different P5 systems may share the same penetration even though the resolution is different due to different levels of turbulence, whereas transmission is expected to vary with the system. I recommend the authors follow the review by Stozenburg (2018) when reporting the DMA parameters, as least reporting penetration together with transmission.

Related to this, Fig. 4 is a bit confusing as none of these DMA peaks reach exactly 1.0 due to diffusional broadening, yet it is probably ok since Fig. 4 is on resolution.

We thank the reviewer for this comment. During the transmission experiments, the voltage for Half Mini DMA was fixed and the voltage for the DMA P5 was scanned continuously. The transmission reported is the maximum transmission efficiency calculated based on the ratio of the peak aerosol concentration recorded by the downstream and mean aerosol concentration recorded by the upstream electrometers. We agree with the reviewer that penetration efficiency and transfer function is the necessary to invert the signal to cluster/aerosol size distribution. However, to our best knowledge, there is no reported transfer function for planar DMA. The review by Stozenburg (2018) only provide the recommended transfer function of diffusing particles for cylindrical DMAs and radial DMAs. We are now working on the derivation of the transfer function for planar DMAs. Considering the time consumption, we want to report only the maximum transmission efficiency in this paper, which, to the best of our knowledge has only been reported once. The reported the transmission of planar DMA P4 (former version of P5) with obtained with similar TDMA system to this work. Compared with the precedent, our work provided more detailed information about how to set up TDMA system for characterizing the ion transmission for planar DMAs, and provided the updated transmission value for the latest version of the planar DMA. In.the revised manuscript, we have added description of the transmission, which is the maximum transmission efficiency, defined

as the peak height of the ratio of downstream electrometer signal to upstream electrometer signal. We have also included more characterization results of ion transmission in the SI.

The revised contents are shown as following: "***The transmission illustrated is the maximum transmission efficiency, which is defined as the peak height of the ratio of the aerosol number counted by the downstream and upstream electrometer ($N_{down}/N_{up}$). The experimental results of $N_{down}/N_{up}$ under different working condition is shown in Fig. S7 and S8. ……..***". (Line 308-310)

Fig 4 is to compare the sizing resolution of DMA P5 with other cylindrical DMAs, all peak intensity in normalized with the peak height. We have also added description in the figure caption.

The revised contents are shown as following: "***Fig. 4 Comparison of the resolving power of DMA P5 and other commercial DMAs for detecting THA+. DMA P5 was operated under the sheath flow rate of 1500 L/min, Half Mini was operated under the sheath flow rate of 300 L/min. All the signal intensiy is normalized with the peak height.***" (Line 225)

d. More discussion is needed in Section 3.2 and some parts in 3.1, including more details on different modes of the setup and the applicability of the results to other setups. For instance, when measuring atmospheric clusters, the clusters are introduced to the DMA from the polydispersed aerosol flow inlet, right? Then the discussion related to Fig. 1 can be confusing as clusters are injected into the DMA using an electrospray. Whether electrostatic losses vary with the working modes also needs to be explained.

We thank the reviewer for this comment. We have added more discussion in section 3.2 about the application of TDMA system for the characterizing the ion transmission efficiency. And more details on setup in both Section 3.1 and Section 3.2. The discussion about Fig.1 is about how we use standard ions generated from electrospray source to characterize the DMA P5. We agree with the reviewer that we have not included the explanation of how the DMA P5 works for measuring the atmospheric clusters. When measuring atmospheric clusters: (1) Under injection mode, the clusters are introduced to the DMA from the polydispersed aerosol flow inlet. The reagent ions are generated via electrospraying of custom selected solutions. The regent ions charge the oxidation products through secondary electrospray ionization (SESI) (Rioseras et al., 2017). (2) Under counter flow mode, the blocked port (red labeled in Fig. 1b) is used to introduce the atmospheric clusters, while the $O_{count}$ is equal to the sum of the counter flow rate and the sample flow rate. In the revised

manuscript, we have added the explanation of how we set up DMA P5 for the measurement of atmospheric clusters in section 3.1. The revised sentences are shown as following:

1. "*The key point of applying ESI source with DMA P5 is that the ESI voltage is not constant under scan mode. The voltage applied to the nano-ESI source floats above the inlet electrode voltage of DMA P5 with the floating value being the exact ESI voltage.*" (Line 131-133)

2. "*According to such design, as being illustrated in the former section, two recirculation modes can be applied for DMA P5 operation for characterizing its performance with standard aerosol generated from nano-ESI source.*" (Line 135-136)

3. "*The performance of DMA P5 for obtaining the THA+ mobility spectrum under suction mode is shown in Figure S3. With solution concentration higher than 0.5mM, well separated THA+ monomer, dimer and trimer can be observed. Hihger solution concentration leads to the increase of the signal/noise ratio, but decreases the dimer/monomer ratio. As can be derived from Eq. (1) and Eq. (4), both the VDMA and corresponding sizing resolution of THA+ increases with the increase of sheath flow rate.*" (Line 153-157)

4. "*It should also be noted the above-mentioned recirculation set ups are applied for the study of aerosols generated from ESI source. For the measurement of atmospheric clusters, secondary electrospray ionization (SESI) (Rioseras et al., 2017) is applied, with the reagent ions generated via electrospraying of custom selected solutions. Under injection mode, the clusters are introduced to the DMA from the polydispersed aerosol flow inlet. Under counter flow mode, the blocked port (red labeled in Fig. 1b) is used to introduce the atmospheric clusters, while the Ocount is equal to the sum of the counter flow rate and the sample flow rate.  Gao et al. (2023) have deployed the SESI-DMA-TOF for the measurement of the products of α-pinene ozonolysis.*" (Line 197-202)

Reference

*Rioseras, A.T., Gaugg, M.T., Martinez-Lozano Sinues, P.: Secondary electrospray ionization proceeds via gas-phase chemical ionization. Anal. Methods 9, 5052-5057, 2017.*
*Gao, J., Xu, Z., Cai, R., Skyttä, A., Nie, W., Gong, X., Zhu, L., Cui, S., Pei, X., Kuang, B., Kangasluoma, J., Wang, Z.: Molecular identification of organic acid molecules from α-pinene ozonolysis. Atmospheric Environment, 312, 2023.*

e. The abstract and conclusion could be sharpened such that the audience can better understand the main contribution of this study to the research community. For instance, "we assessed the performance of a commercial planar DMA integrated with the home-build recirculation system" in the abstract is rather confusing, as it seems to emphasize that the results of this study do not apply to other P5 DMA systems due to the difference in the recirculation system.

We thank the reviewer for this comment. The original intension of mention the home-build recirculation system is that all components can be purchased from the local market. Our reported transmission and resolution are expected to be reproduced under suction mode or counter flow mode of DMA P5 with any recirculation system providing temperature-constant, aerosol-free, stable sheath flow. We have re-arranged the abstract and conclusion in the revised manuscript, leaving out the emphasis of the home-made recirculation system. The revised sentences are shown as following:

Abstract: "*The planar differential mobility analyzer (DMA) serving as particle sizer can achieve higher transmission and selection precision at ambient pressure compared with conventional cylindrical DMA, and show potentials on coupling with atmospheric pressure interface mass spectrometer (API-MS) for cluster detection with an additional ion mobility dimension. In this study, we assessed the performance of a commercial planar DMA (DMA P5). The sizing range of the system in this work is sub-3.9 nm, although larger sizes can be measured with a sheath gas flow restrictor. The resolving power under different recirculation setups (suction mode and counterflow mode) and different sheath flow rates was evaluated using electrosprayed tetra-alkyl ammonium salts. The maximum resolving power of THA+ under suction and counterflow mode are 61.6 and 84.6, respectively. The sizing resolution of DMA P5 is 5-16 times higher than conventional cylindrical DMAs. The resolving power showed approximately linear correlation with $\sqrt{V\_DMA}$ under counterflow mode, while the resolving power of THA+ under suction mode stopped linearly increasing with $\sqrt{V\_DMA}$ when the VDMA was above 3554.3V and entered a plateau due to the interference of sample flow on the laminarity of sheath flow. The transmission efficiency of DMA P5 can reach 54.3%, about one order of magnitude higher than the commercial DMAs. The mobility spectrum of different electrosprayed tetra-alkyl ammonium salts and the mass to charge ratio-mobility 2D spectrum of sulfuric acid clusters was also characterized with the DMA P5 (-MS) system.*"

Conclusion: "*We present the characterization results of a planar DMA (DMA P5). The sizing range of DMA P5 is sub 3.9 nm. Two operation modes can be applied (suction mode and counter flow mode). Under suction mode, the maximum resolving power can reach 60, while under counter flow mode, the maximum resolving power is 84. The resolving power of DMA P5 can be 5-16 times higher than commercial DMAs. Under suction mode, the obtained monodispersed aerosol number concentration can be modified by changing the injection flow rate. Under counter flow mode, although the resolving power is higher than the suction mode, the obtained monodispersed aerosol number concentration is lower due to the absence of injection flow. The ion transmission of DMA P5, tested by a TDMA system, exceeds 54.3%, which is about 7-8 times higher than commercial DMA (TSI 3086).*

*The application of DMA P5 was also characterized. Positive and negative aerosol ions of four tetra alkyl ammonium halides (THAB, TMAI, TBAI and TDAB) were measured, and high-resolution ion spectra were obtained. Finally, P5 was combined with an API-TOF-MS to successfully measure the two-dimensional (mass to charge ratio V.S. ion mobility) distribution of sulfuric acid clusters. The mobility diameters of sulfuric acid clusters (monomer to tetramer) were measured.*

*This system can be used to simultaneously measure the ion mobility and chemical composition of atmospheric clusters. In addition, this system can also be applied to calibrate the mass dependent ion transmission efficiency of mass spectrometry and study the impact of the collision induced cluster fragmentation (CICF) inside the mass spectrometry on the measurement results of atmospheric clusters.*"

2. Due to the lack of explanation, the comparison among different DMAs seems to be a bit arbitrary.
a. Taking Fig. 4 as an example, what are the sheath and aerosol flow rates of the DMAs, and why these values are used for comparison? Were the aerosol-to-sheath flow rates are the same for all the DMAs or the aerosol flow rate is set to the same value? Are the flow rates chosen to represent typical ambient measurement conditions or they are for different conditions? The underlying question behind these several questions is, if e.g. TSI 3086 DMA works with a resolution of 5 in a setup for ambient cluster/particle measurement, will the P5 DMA provide a resolution higher than 50, or does it simply not usable due to the difference in flow configurations?

We thank the reviewer for this comment. We have added the explanation of Fig. 4, reporting the corresponding sheath and aerosol flow rates of the DMAs. The aerosol-to-sheath flow ratio for all reported cylindrical DMAs (except HalfMini DMA) is approximately 10, which is the typical flow configuration for particle sizing in both lab and field measurements. The aim of making this comparison is that although application of DMA P5 on atmospheric particle number size distribution measurements is unpractical due to the high maintaining expenses for keeping the super high sheath flow rate, the exceptional sizing resolution and high ion transmission of DMA P5 is merit of being further exploited by coupling with mass spectrometer for cluster detection with an additional ion mobility dimension.

The added contents is shown as following: "…..*The DMA P5 was operated under counter flow mode at the sheath flow rate of about 1500 L/min (corresponding to the Vblower of 8.5 V). The Half Mini DMA was operated at the aerosol-to-sheath flow ratio of 10/300 L/min. The reported resolution was measured under the aerosol-to-sheath flow ratio of 0.6/6 L/min for the Caltech nanoRDMA, of 6/61.4 L/min for the Vienna DMA, of 2/21.9 L/min for the Grimm nanoDMA, of 2.0/20 L/min for TSI 3085, of 2.5/25 L/min for TSI 3086 and of 1.5/15 L/min for the Caltech RDMA. The aerosol-to-sheath flow ratio for all reported cylindrical DMAs (except HalfMini DMA) is approximately 10, which is the typical flow configuration for particle sizing in both lab and field measurements. The comparison results show that the planar DMA has the highest sizing resolution, which is 5-16 times higher than conventional cylindrical DMAs (Fig. 4). On one hand, due to the high maintaining expenses for keeping the super high sheath flow rate, application of DMA P5 on atmospheric particle number size distribution measurements is unpractical. On the other hand, high resolution and high ion transmission are almost synonymous for planar DMAs. This advantage is merit of being further exploited by coupling with mass spectrometer for cluster detection with an additional ion mobility dimension…....*". (Line 209-220)*

b. Related to this, it seems the "conventional cylindrical DMAs" in the abstract (line 21) do not include Hermann DMA and the half-mini DMAs. Why?

We thank the reviewer for this comment. Hermann DMA and the half-mini DMAs are conventional cylindrical DMAs. We have corrected the comparison results of the sizing resolution between DMA P5 and conventional cylindrical DMAs.

The revised sentences are shown as following: "***The sizing resolution of DMA P5 is 5-16 times higher than conventional cylindrical DMAs.***"

c. The results in Fig. 5b need to be improved. TSI 3086 was not developed in 2011 so one cannot only cite Jiang et al. without explanation. It might be better to use the same bar for TSI 3085 and 3086 and cite Stolzenburg et al. (2018). The transmission of the Grimm nanoDMA has been improved and the results can be found in Stolzenburg et al. (2017).

We thank the reviewer for this comment. We have improved the results and citation in Fig. 5b (Figure 8b in the revised manuscript), which is shown below:

[Figure]

**Fig. 8 (a) Ion transmission efficiency of P5 under different Qout; (b) Comparison with other cylindrical DMAs (the red bars represent the experimental results).**

d. The resolution in Fig. 2a looks different from Fig. 3 in Amo-González and Pérez (2018). I would like to see a discussion on this difference and how it affects the results.

We thank the reviewer for this comment. The theoretical resolution of DMA P5 is different from the calculation in Amo-González and Pérez (2018). Dr. Amo-González, as the co-author of this paper, have pointed out that there was a mistake on the formulas in his published paper. We have used the corrected formula in this paper, which showed higher theoretical value of sizing resolution. The detailed derivation of Eq. (4) has been added in the SI (Section 3).

3. The novelty of the study can be better emphasized by shortening Section 3.3 (moving some parts to the SI) and leaving more space for 3.1 and 3.2. Fig. 8 can be moved to Section 3.1. The authors are encouraged to emphasize more on the characterization results instead of emphasizing the high resolution of P5 without restricting the working conditions, as it is known that P5 can reach a high resolution of > 100 at certain conditions.

We thank the reviewer for this comment. In the revised, we have added more discussions in section 3.1 and 3.2, emphasizing the characterization results of P5. Giving more details about how to set up DMA P5 system for characterizing its performance with standard ions generated from electrospray sources and for conducting experiments for characterizing transmission efficiency through TDMA system. We think the revised version is better organized for the reader to follow. On the other hand, the advantage of DMA P5, compared to cylindrical DMAs, is its exceptional ion transmission and the capability of coupling with MS. With the current setup, this technique can be very useful in laboratory studies of atmospherically relevant clusters, we have also added more details about the interpretation of the 2D spectrum of sulfuric acid clusters. The revised contented is shown below:

1. "*The key point of applying ESI source with DMA P5 is that the ESI voltage is not constant under scan mode. The voltage applied to the nano-ESI source floats above the inlet electrode voltage of DMA P5 with the floating value being the exact ESI voltage.*" (Line 131-133)

2. the characterization results of three other tetra alkyl ammonium halides have been moved to section 3.2 in the revised manuscript (Line 227-249)

[revised manuscript text omitted]
 towards atmospheric cluster measurements". It seems none of the measured clusters in this study are sampled from the atmosphere.

We thank the reviewer for this comment. Our measured clusters were not sampled from the really atmosphere, but generated by electrospray. These clusters have the same (or similar) element composition and physicochemical properties with the atmospheric clusters. We agree with the reviewer that the current title cannot precisely reflect the content of our experiment. We have changed our title to "*Characterization of the planar differential mobility analyzer (DMA P5):resolving power, transmission efficiency and its application to atmospheric relevant cluster measurements*" in the revised manuscript.

5. The P5 DMA is described as newly developed, which is a bit confusing. Is it a new model or the same as the one reported by Amo-González and Pérez (2018)? Also, the DEG-SMPS in 2011 cannot be described as "newly developed".

We thank the reviewer for this comment. We agree that newly developed is not an appropriate description for both DMA P5 and DEG-SMPS. In the revised manuscript, we have removed "*newly developed*" for both DMA P5 and DEG-SMPS.

6. a. line 17, page 1, abstract, "0-3.9 nm". Better to use sub-3.9 or start with a very small diameter. 0 nm does not practically make sense.

b. line 23, page 1, abstract, "stopped linearly increase". increasing?

c. line 23, page 1, abstract, "enter a plateau". Entered

d. line 24, page 1, abstract, "one factor of magnitude". one order of magnitude or a factor of 10

e. "thorough" in multiple places. through.

f. line 149, page 6, "much closer". Significantly closer. It is still far from the ideal resolution.

g. Table 1, diameter. Please specify which diameter it is in the caption or the table header.

h. line 278, page 14. "is 7-16 times higher". Can be. "is" is too strong and hence incorrect as it depends on the flow configurations.

i. Please check the colon in the title

We thank the reviewer for the comment. All the above-mentioned inappropriate words or grammar errors have been corrected.

References:

Stolzenburg, D., Steiner, G., and Winkler, P. M.: A DMA-train for precision measurement of sub-10 nm aerosol dynamics, Atmospheric Measurement Techniques, 10, 1639-1651, 10.5194/amt-10-1639-2017, 2017.

Stolzenburg, M. R.: A review of transfer theory and characterization of measured performance for differential mobility analyzers, Aerosol Science and Technology, online available, 10.1080/02786826.2018.1514101, 2018.

Stolzenburg, M. R., Scheckman, J. H. T., Attoui, M., Han, H.-S., and McMurry, P. H.: Characterization of the TSI Model 3086 Differential Mobility Analyzer for Classifying Aerosols down to 1 nm, Aerosol Science and Technology, 52, 748-756, 10.1080/02786826.2018.1456649, 2018.

Amo-González, M. and Pérez S.: Planar Differential Mobility Analyzer with a Resolving Power of 110, Analytical Chemistry, 90, 6735–6741, 10.1021/acs.analchem.8b00579, 2018

**Referee #2**

This work reports characterization of a parallel plate DMA (P5). The resolving power and transmission efficiency of the system are measured at different instrument operating conditions and the reasons behind their variations are discussed. Afterwards the DMA is used to characterized sulfuric acid clusters, demonstrating its potential application to atmospheric clusters. This work falls into the scope of AMT and it may be published after major revisions.

**Major comments:**

1. Line 17: Can the sizing range reach 0? No DMA can size infinitely small particles (e.g., electrons) due to diffusion. Even for ions I believe there is some limit if the size of the ion gets very small.

We thank the reviewer for the comment. We agree with the reviewer that the lower sizing limit of DMA sizing can never reach 0. We have replaced the sizing range to "***sub-3.9nm***" in the revised manuscript.

2. Line 23: 'when the $V_{DMA}$ was above 3554.3V'. It is more appropriate to report a flowrate here (and in other similar sentences in the manuscript) since the authors have argued it is a flow field effect that affect the system resolution.

We thank the reviewer for the comment. The sheath gas flowrate/velocity is one of the key parameters for DMA P5 to achieve high sizing resolution. However, the flowrate of DMA P5 is too high for precise measurement. $V_{DMA}$ is constantly monitored and logged, and is tightly connected with sheath gas flowrate and the sizing resolution (R). As can be derived from Eq. (1) $Z = \frac{U \cdot h^2}{L \cdot V_{DMA}}$, the ratio of $V_{DMA}/U$ (sheath flow velocity in the separation region) is a constant value for aerosol with fixed ion mobility. As can be derived from Eq. (4), the R variation of is dependent on the change of $\sqrt{V_{DMA}}$. Consequently, we hope to report $V_{DMA}$ in the manuscript. Estimation of sheath flow rate in the symmetry plane going through the center of the inlet slit with different $V_{blower}$ is shown in Fig. S2.

3. Line 166: What is the reason that higher Qin leads to higher signal strength? Are more ions carried to the DMA inlet by the higher flowrate.

We thank the reviewer for the comment. We think the main reason that higher $Q_{in}$ leads to the higher signal strength is the decreased diffusion loss. The mobility diameter of THA$^+$ (ions used for

characterizing the performance of DMA P5) is 1.47 nm. The diffusive losses are not negligible for these small ions. Higher $Q_{in}$ can decrease the retention time of ions in the nano chamber, and increase the number of ions reaching the DMA inlet.

4. In Figure 2b, I suppose there should be two lines of the counter-flow mode curve corresponding to Qout = 1L/min and Qout = 2L/min.

We thank the reviewer for the comment. We have added the curve of counter-flow mode under Qout = 1L/min in Fig. 2b in the revised manuscript, which is shown below:

[Figure]

Fig 2. (a) The dependency of the resolving power of THA$^+$ on DMA voltage ($V_{DMA}$) under suction mode and counter flow mode; (b) dependency of the resolving power and signal intensity on the $Q_{out}$ under suction mode and the comparison with counterflow mode.

5. The phrase 'signal intensity' is bit ambiguous in the manuscript. In line 162, it seems to refer to the 'number concentration of the sizing aerosol'. In Line 170, it refers to the total current measured by the electrometer. Please make it clear what signal intensity means exactly throughout the manuscript.

We thank the reviewer for the comment. The output signal of the electrometer is in the unit of Volt (V) and the signal intensity range is 0-2V. The amplification value of the electrometer is $10^{11}$ V/A for Lynx E11 and $10^{12}$ V/A for Lynx E12. With the amplification value the output signal can be converted to the actual current intensity. The current intensity can be further converted to number concentration with known flow rate and net charge of the measured ions. We have added the explanation of how the raw output data (in V) convert to ion current (pA) in section 2.

The revised part is shown as following: "......*The Faraday cage electrometers (Lynx E11&E12, SEADM, Valladolid, Spain, Fernandez de la Mora et al., 2017) were used as particle counter. The*

*output signal range was 0-2V, with an amplification of $10^{11}$ V/A and $10^{12}$ V/A, respectively. ……*".(Line 107-109)

Moreover, we have changed the total current to the electrometer output unit (in V) in the revised manuscript (Fig.3), unifying the signal intensity as the direct output value to the electrometer throughout the manuscript. The updated Fig.3 is shown below:

[Figure]

**Fig. 3 Mobility spectrum of THA$^+$ under different $Q_{out}$ with (a) $Q_{counter}$ = 0.5 L/min; (b) $Q_{counter}$ = 1.0 L/min; (c) Integrated signal intensity, resolving power and V$_{DMA}$ of THA$^+$ under different $Q_{out}$.**

6.  Are the lines in Fig. 4 measured/calculated/taken from literature?

We thank the reviewer for the comment. The dashed lines in Fig 4 are measured with our P5 and HalfMini DMA in our laboratory, while other lines for the commercial DMAs are taken from the literature. In the revised manuscript, we have updated of the commercial DMAs and added explanation of where we cited these values. The added contents are shown as following "*……The sizing resolution of THA$^+$ monomer by DMA P5 and Half Mini DMA (Fernandez de la Mora and Kozlowski, 2013), measured in our lab, were compared with the reported results of different types of commercial DMAs (Jiang et al., 2011, Stolzenburg et al., 2018). The DMA P5 was operated*

*under counter flow mode at the sheath flow rate of about 1500 L/min (corresponding to the Vblower of 8.5 V). The Half Mini DMA was operated at the aerosol-to-sheath flow ratio of 10/300 L/min. The reported resolution was measured under the aerosol-to-sheath flow ratio of 0.6/6 L/min for the Caltech nanoRDMA, of 6/61.4 L/min for the Vienna DMA, of 2/21.9 L/min for the Grimm nanoDMA, of 2.0/20 L/min for TSI 3085, of 2.5/25 L/min for TSI 3086 and of 1.5/15 L/min for the Caltech RDMA. The aerosol-to-sheath flow ratio for all reported cylindrical DMAs (except HalfMini DMA) is approximately 10, which is the typical flow configuration for particle sizing in both lab and field measurements.* ...” (Line 207-215)

7. Section 3.2:The P5 was operated at fixed voltages corresponding to the THA+ monomer peak?

We thank the reviewer for the comment. When applying the TDMA system for ion transmission measurement, the upstream Half Mini DMA was operated at fixed voltage corresponding to the THA$^+$ monomer peak. Downstream to the Half Mini DMA, the monodispersed THA$^+$ monomer passed through a flow splitter, reaching DMA P5 and the first electrometer. The DMA P5 was operated under scan mode and was connected to the second electrometer, to obtain the full mobility spectrum of THA$^+$ monomer. We have added the description of DMA P5 operation when characterizing the transmission efficiency in the revised manuscript. As well as detailed discussion in Section 3.2.

The description of the operation status of DMA P5 is shown as following: “… *During the experiments, the voltage for Half Mini DMA was fixed and the voltage for the DMA P5 was scanned continuously.* …”. (Line 292-293)

8. Fig 5a: It is interesting to know if there is an upper limit for the positive relation between ion transmission and Qout.

We thank the reviewer for the comment. We believe that there is an upper limit for the positive relation between ion transmission and Qout. Higher Qout can not only compensate the electrical velocity generated from the electric field between outlet electrode with a high negative voltage and the grounded electrometer, but also decrease the diffusive losses from the aerosol beam splitter to the inlet slit at the inlet electrode of DMA P5. The reason we do not try higher Qout is that 3 L/min is a quite high value with respect to the geometry of exit slit (1.0mm in diameter) at the exit electrode and the inlet slit (0.6mm width, 7mm length). Since the original outlet slit of the parallel plate DMA was designed to be coupled to the vacuum system of a mass spectrometer. Based on our current

experimental condition, it is hard to obtain Qout large enough to find to turning pointing, after which the relation of ion transmission and Qout reaches the plateau. It should be noted that under conventional DMA P5 configuration (both outlet electrode and detectors are grounded), the ion loss due to the electrical dragging force is negligible. Our results represented the lower limit of the DMA P5 ion transmission efficiency operated under conventional configuration. This lower limit value (54.3%) is 4.5-17.5 times higher than other commercial cylindrical DMAs.

9.  Fig 5a: Another interesting comparison would be comparing the transmission of ions with different sizes at the same flowrate (using ions presented in Fig 6). It would be interesting to know if a single transmission can be applied to different ions at a given flow configuration.

We thank the reviewer for the comment. During the experiments, the voltage for Half Mini DMA was fixed and the voltage for the DMA P5 was scanned continuously. The transmission efficiency reported in this study is the maximum ratio of the aerosol concentrations recorded by the downstream and upstream electrometers. To the best of our knowledge, there was only one paper, reporting the transmission of planar DMA P4 (former version of P5). The reported value was about 50%. Our results indicated that the lower limit transmission of DMA P5 was ~5% higher than its former version. It should also be noted that the characterization of transmission of DMA P4 and other cylindrical DMAs used $THA^+$ as standard ions, due to the intensively studied ion mobility and the capability of generating monodispersed $THA^+$ monomer. The reason we reported only $THA^+$ is to compare the performance of DMA P5 with other DMAs. The exceptional transmission indicates that DMA P5 deserved to be further exploited for atmospheric cluster studies by coupling with MS. The combination of DMA P5 with API-TOF-MS shows that it is already a useful tool in the laboratory studies of atmospherically relevant clusters.

We agree with the reviewer that it is interesting to conduct transmission characterization for different tetra alkyl ammonium halides. We think the transmission of different ions at a given flow configuration is different, because of the different effect of diffusion broadening for ions with different mobility. Clarifying the transmission of different ions in DMA P5 needs not only further experimental studies, but also theoretical studies of the transfer function of planar DMA, which, to our best knowledge, have not been reported before. Consequently, the transmission of different tetra alkyl ammonium halides was not studied in this paper.

10.  Atmospheric clusters -> atmospherically relevant clusters. For the DMA-electrometer or DMA-

MS system, one challenge to detect the atmospheric clusters is their low concentration. It has not been shown that atmospheric clusters can actually be measured by the parallel plate DMAs in this manuscript.

We thank the reviewer for the comment. Our measured clusters were not sampled from the real atmosphere, but generated by electrospray. These clusters have the same (or similar) element composition and physicochemical properties with the atmospheric clusters. We agree with the reviewer that the current title cannot precisely reflect the content of our experiment. We have changed our title to "***Characterization of the planar differential mobility analyzer (DMA P5):resolving power, transmission efficiency and its application to atmospheric relevant cluster measurements***" in the revised manuscript. Though the detect limit of our system need to be further evaluate for ambient measurement, it, with its current form, can be a good tool for studying the physicochemical properties of atmospherically relevant clusters in the lab.

**Technical corrections:**

Line 58: parallel plate

We thank the reviewer for the comment. We have changed the "parallel electrodes" to "parallel plates" in the revised manuscript.

Line 94: springer?

We thank the reviewer for pointing out this spelling mistake. We have corrected the "springer" to "syringe" in the revised manuscript.

Eq. (4): what is delta_L0.5?

We thank the reviewer for the comment. The DMA sizing resolution is defined as the mean ion mobility divided by the full mobility width at half-maximum (fwhm). In $\frac{\Delta L_{0.5}}{L}$, $\Delta L_{0.5}$ represents fwhm, L represents the mean mobility. To avoiding misunderstanding, we have modified Eq. (4) in the revised manuscript, following the expression of Eq. (3).

**Referee #3**

Discussion of: **Characterization of the planar differential mobility analyzer (DMA P5): resolving power, transmission efficiency and its application to atmospheric cluster measurements**, by: Zhengning Xu, Jian Gao, Zhuanghao Xu, Michel Attoui, Xiangyu Pei, Mario Amo-González, Kewei Zhang, Zhibin Wang

by Juan Fernandez de la Mora, Yale University, Mechanical Engineering Department.

The article describes an instrument combination previously used in laboratory studies, but, to my knowledge, not in atmospheric studies. I see merit in that approach, and consider this exploratory article a valuable contribution to the field of atmospheric measurements. It covers prior work fairly, and adheres to high scientific standards. I support its publication, pending some desirable improvements. There is the important issue of whether the instrument will be sufficiently sensitive for relevant atmospheric studies. Whether or not this is the case, the article already shows clearly that the instrument can be very fruitful at least in laboratory studies of atmospherically relevant clusters. In this I believe there are precedents that ought to be cited.

**1.Abstract.**

It would be preferable to avoid the abbreviations TMAI, TBAI, THAB and TDAB. Also the term "newly developed" seems inappropriate for an instrument that has existed for considerable time.

We thank the reviewer for the comment. We have removed the abbreviations in the abstract in the revised manuscript. We agreed with the reviewer that it is inappropriate to described DMA P5 as newly developed, since Amo-González et al. have coupled DMA P5 with mass spectrometer in 2018. We have removed the "*newly developed*" in the revised manuscript.


The transmission study is most valuable, as I am not aware of prior quantitative studies of this

important metric for planar DMAs that would be directly relevant to atmospheric measurements. There are studies with electrospray sources directly facing a planar DMA inlet slit, showing many orders of magnitude gains in transmitted ion signal versus cylindrical DMAs. However, an electrospray is an intense unipolar ion source injecting several hundred nA of current into an area typically smaller than $1\mu m^2$. In this case, the major source of ion loss is beam broadening by space charge, which would not be relevant in most atmospheric measurements. In a planar DMA the electrospraying capillary can be brought arbitrarily close to the inlet slit. Under such conditions, Javaheri et al. (2008) have shown that almost all molecules of an electrosprayed dilute species in solution can be transmitted as ions through the atmospheric inlet orifice of a mass spectrometer. Similarly, over 1 nA of the dominant electrosprayed ion can be transmitted to the outlet slit of a planar DMA (Fernandez de la Mora, 2019). This exceptional transmission has been exploited by Tauber et al. (2018) to introduce DMA-purified highly concentrated atomic ions into a nucleation chamber to study ion induced nucleation. Their study suggests that the same is not possible with any cylindrical DMA. Similarly, in our first tandem DMA-MS study involving a Time of Flight (TOF) mass spectrometer, it took us over an hour with a high resolution cylindrical DMA (passing 10 L/min of aerosol) set at a fixed voltage to accumulate a single mass spectrum with useful information on (THABr)n(THA+)z clusters (Fernandez de la Mora et al. 2005). This is the main reason why most successful couplings of DMAs with MS systems (mainly using electrospray sources) have involved planar DMAs. There are nevertheless exceptions involving quadrupole MS systems set at a fixed mass, while a cylindrical DMA scans over the voltage (Ude et al., 2004). Steiner et al. (2014) have successfully coupled a cylindrical DMA of relatively high resolution to a TOF-MS to investigate small ions from radioactive bipolar sources.

The present transmission study uses a tandem setup with a cylindrical first DMA. In this case the vast majority of electrospray ions are lost by space charge in the aerosol inlet tube and in the annular region preceding the inlet slit to the analyzing region of the first (cylindrical) DMA. Consequently, there is little space charge left when these ions reach the second (planar) DMA. Therefore, the losses measured by Xu et al. are primarily diffusive losses, which is what would be relevant in atmospheric sampling. These losses are not negligible for the small ions investigated by Xu et al., and the observed substantial advantage of planar over cylindrical DMAs (Figure 5b) is apparently also due to lack of an extended annular region upstream of the injection slit. This dominant region of diffusive

losses is evidently reduced in cylindrical DMAs using a small outer radius R2, which explains the advantage of the Half-Mini DMA (R2=7 mm) reported in figure 5b over other cylindrical DMAs. It would be useful if the authors would report the geometry in the cubic chamber used in the planar DMA upstream of the injection slit, since this might be the major source of the observed 46% ion loss. Most relevant to these losses is how far from the slit is the end of the tube bringing in the aerosol into this cubic chamber.

We thank the reviewer for the comment. The reviewer has listed numbers of outstanding works from the precedents about the ion transmission study of planar DMA and the comparison with cylindrical DMAs. Moreover, the reviewer also well illustrated the benefit of applying TDMA on studying the ion transmission of DMA P5 for atmospheric relevant study. We have cited these works in the revised manuscript.

Moreover, as being pointed out, the geometry of cubic chamber is related to the reported ion transmission efficiency. The inner geometry of cubic chamber was cylinder-shaped, with the diameter of 32 mm, and the height of 6.5 mm. A straight conductive silicone tube is inserted into the center of the cylinder-shaped space thorough the hole on the top of the chamber. The space between the conductive silicone tube and the hole is sealed with silicone glue. The injection end of the tube is connected to the beam splitter, while the exit end is at the same plane with the lower end of the chamber. When the chamber is sealed to the inlet electrode of DMA P5, the distance between the exit end of tube and the inlet slit of DMA P5 is about 5.2 mm. In the revised manuscript, the geometry of cubic chamber was reported. The detailed description of how the monodispersed THA$^+$ ions are injected to the inlet slit was also added in the revised manuscript.

The added sentences are shown as following: "….. *Downstream to the Half Mini DMA, DMA P5 operated at scan mode was connected between the flow splitter and the second electrometer. A 3D-printed cubic chamber was used was used to connect the inlet tube and DMA inlet slit. The inner geometry of cubic chamber was cylinder-shaped, with the diameter of 32 mm, and the height of 6.5 mm. The inlet tube was inserted straightly into the centre of the cylinder-shaped space thorough the hole on the top of the cubic chamber. The space between the conductive silicone tube and the hole was sealed with silicone glue. The injection end of the inlet tube was connected to the beam splitter, while the exit end was at the same surface with the lower end of the chamber. When the cubic chamber was sealed to the inlet electrode of DMA P5, the distance*

*between the exit end of inlet tube and the inlet slit of DMA P5 is about 5.2 mm.* ….." (Line 286-292). We hope other group using DMA P5 can find useful information from our work.

**3. Mass spectrometer selection**

Xu et al. use a TOFWORK AG mass spectrometer. In this they follow the lead of many widely cited atmospheric studies by Kulmala and colleagues. However, most other past DMA-MS couplings have relied on other commercial time of flight mass spectrometers developed broadly for electrospray mass spectrometry studies, many of them having much higher resolving power and mass range than the TOFWORK MS. These other instruments have achieved high reliability and ion transmission efficiency, and would at first sight seem to be ideally fitted for coupling with a DMA for atmospheric studies. It would be of considerable interest to those intending to pursue related atmospheric DMA-MS studies to learn about the considerations that have led Xu et al. to their MS choice.

We thank the reviewer for the comment. There were two main reasons that we chose the APi-TOF from TOFWORK AG to couple with DMA P5 for atmospheric relevant clusters study: *(1) Low detection limit*: the APi-TOF consists of a time-of-flight mass spectrometer (TOF) coupled to an atmospheric pressure interface (APi). The Api has three differentially pumped chambers, the first two containing short segmented quadrupoles used in ion guide mode, and the third containing an ion lens assembly. The APi-TOF has a very low background noise level and detection limit. Junninen et al., (2010) reported that the detection limit of the APi-TOF within the mass range of 80-900 was below 1 ion/cm$^3$. Taking the high ion transmission of DMA P5 into consideration, we think coupling DMA P5 with the APi-TOF have the potential for the measurement of atmospheric clusters. Moreover, both Api-TOF and Api-TOF coupling with a chemical ionization source (CI-API-TOF) have been successfully applied for atmospheric cluster measurement in laboratory and field, exhibiting the capability of detecting different atmospheric clusters, such as cluster complexes of sulfuric acid with ammonia (Kirkby et al., 2011; Lehtipalo et al., 2016), amine (Almeida et al., 2013; Yin et al., 2021), and organics (Riccobono et al., 2014). *(2)* **Adjustable APi configuration:** Fragmentation of molecular clusters inside MS is a significant source of uncertainty in a wide range of chemical applications. Different clusters fragmented differently inside MS due to the different binding energy (*Lopez-Hilfiker et al., 2016; Passananti et al., 2019*). The voltage configuration, combination of the voltages applied to the APi-TOF, can significantly affects the transmission and

fragmentation of clusters. By the combination of API-TOF and DMA-P5, we aim to study the physiochemical properties of the selected atmospheric (relevant) clusters (based on their ion mobility) by adjusting the voltage configuration, scanning the electric field strength within the transfer optics in real time while measuring a steady-state distribution of the selected clusters. With this function, we can experimentally determine the electric field strength required to break apart atmospheric (relevant) clusters, which are directly related to the binding energy of the clusters. The related experimental results are not within the scope of this work, and will be reported in another paper, currently under preparation. In the revised manuscript, we have added the consideration of why we choose API-TOF to couple with DMA P5.

The revised sentences are shown as following:

1. "*There are excellent commercial instruments able to measure both mobility and mass in tandem (ion mobility spectrometry-mass spectrometer, IMS-MS). Most of them use either intense potentially fragmenting electric fields in the mobility analyzer, or carry the mobility analysis in a region of reduced pressure (May and McLean, 2015). What is special about the DMA P5 is that it operates at atmospheric pressure and has little tendency to fragment even weakly bound clusters, which is very suitable for the detection of atmospheric clusters.* …" (Line 327-331)

2. "……. *API-ToF-MS has a very low background noise level and detection limit (< 1 ion/cm3, Junninen et al., 2010). Moreover, both Api-TOF and Api-TOF coupling with a chemical ionization source (CI-API-TOF) have been successfully applied for atmospheric cluster measurement in laboratory and field, exhibiting the capability of detecting different atmospheric clusters, such as cluster complexes of sulfuric acid with ammonia (Kirkby et al., 2011; Lehtipalo et al., 2016), amine (Almeida et al., 2013; Yin et al., 2021), and organics (Riccobono et al., 2014). By the combination of API-TOF and DMA-P5, it is also possible to study the physiochemical properties of the atmospheric relevant clusters scanning the electric field strength within the transfer optics (Lopez-Hilfiker et al., 2016).* ……." (Line 335-341)


*Line 244 states "the resolving power of planar DMA is directly related to VDMA and Ne." Does Ne refer to the negative spray? Please clarify the relevance of this, as it is not at all clear.

We thank the reviewer for the comment. Ne represents the net charge of the aerosol. We have added the description of Ne in the revised manuscript.

*Line 165: The program Igor is quoted for mobility peak analysis. Would you please provide a little more background for those unfamiliar with this tool?

We thank the reviewer for the comment. We have added the description of the analysis package used for multipeak fitting. We used Multi-peak Fitting 2 package embedded in Igor program for the peak fitting. This package was designed to provide curve fits to multiple, overlapping peaks of any sort of measurement that results in localized peaks or lines. The program can also give you an estimate of the peak heights, widths, locations and areas.

* The authors note that their recirculation circuit is not part of the commercial system, perhaps to warn readers of the possibility that DMA performance may depend on this component of the system. I doubt that the flow control part will be much effect on DMA performance, though I may be wrong. One original component in this recirculating flow system perhaps deserves some comment. This is the planar commercial HEPA filter, apparently sandwiched between two surfaces with NW-40 connectors. Would the authors please provide some more detail of this design? *

We thank the reviewer for the comment. The original intension of mention the home-build recirculation system is that all components can be purchased from the local market. Our reported transmission and resolution are expected to be reproduced under suction mode or counter flow mode of DMA P5 with any recirculation system providing temperature-constant, aerosol-free, stable sheath flow. We agree with the reviewer that emphasizing home-build recirculation system can lead to the misunderstanding that the reported results may not be able to apply to other P5 DMA systems due to the difference in the recirculation system, which is not our original intention. In the revised manuscript, we have removed the descriptions of "***the home-build***", emphasizing the universal characteristics of the recirculation system that should be applied within DMA P5 system in section.

We have provided more details of the design how we put the HEPA filter into the recirculation in the revised manuscript. The provided description is shown as following "***The particle filter consists a planar commercial HEPA filter (Ref 34230010, Megalem MD143P3, Camfil Farr) and two stainless assembly. Top side of the assembly is a NW40 connector, while the bottom side fits the geometry of the planar HEPA filter. The HEPA filter is sandwiched between the two bottom surfaces of the two assembly, sealed with O-ring and screws.***" (Line 76-80)

The reference to Fernandez de la Mora and Kozlowski given in Figure 5b must be incorrect, as their study did not include transmission measurements. The correct reference must be a later study by Attoui and colleagues.

We thank the reviewer for the comment. We have corrected the reference in the revised manuscript. The reported transmission efficiency for HalfMini (p) is the internal data from SEADM. We have corrected the reference in Figure 5b (Figure 8b in the revised manuscript).

[Figure]

**Fig. 8 (a) Ion transmission efficiency of P5 under different $Q_{out}$; (b) Comparison with other cylindrical DMAs (the red bars represent the experimental results).**

**6. Conflict of interest statement**.

JFM collaborates frequently with authors Michel Attoui and JFM's former graduate student Mario Amo-Gonzalez. JFM and his wife owned half of the now bankrupt company SEADM where Mario Amo Gonzalez led the development of the planar DMA P5. JFM remains keenly interested in the continuation of SEADM's efforts by others, including the company MION SL.